



# An upper limit for slow earthquakes zone: Self-oscillatory behavior through the Hopf bifurcation mechanism from a model of spring-block under lubricated surfaces

Valentina Castellanos-Rodríguez[1], Eric Campos-Cantón[2], Rafael Barboza-Gudiño[3], and Ricardo Femat[4]

[1,3]Instituto de Geología, Universidad Autónoma de San Luis Potosí, México.
[1,2,4]Instituto Potosino de Investigación Científica y Tecnológica A.C., San Luis Potosí, México.

*Correspondence to:* Castellanos-Rodríguez Valentina (valentina@cimat.mx)

**Abstract.** The complex oscillatory behavior of a springblock model is analyzed via the Hopf bifurcation mechanism. The mathematical springblock model is generated by considering the Dieterich-Ruinas's friction law and the Stribeck's effect. The existence of self-sustained oscillations in the transition zone-where slow earthquakes are generated within the frictionally unstable region -is determined. An upper limit for this region is proposed as a function of seismic parameters and frictional coefficients which are concerned with presence of fluids in the system. The importance of the characteristic length scale $L$, the implications of fluids, and the effects of external perturbations in the complex dynamic oscillatory behavior as well as in the stationary solution, are take into consideration.

## 1 Introduction

In the last decade, the study of slow earthquakes (tremors, low and very low frequencies events, and slow slip events) has taken a great relevance because of its possible relationship with the occurrence of large earthquakes. The stress redistribution of slow earthquakes, and the strain in the lowest limit of the seismogenic layer caused by them, could be helpful for a better understanding of the nucleation process of ordinary earthquakes (Ide et al., 2007; Wech and Creager, 2007; Beroza and Ide, 2009; Peng and Gomberg, 2010; Rivet et al., 2011; Ikari et al., 2013; Jolivet et al. , 2013; Abe and Kato, 2014; Audet and Bürgmann, 2011; Bürgmann, 2014; Wallace et al. , 2016). Although the most slow earthquakes have been detected in subduction zones, there are reports of these in other types of faults (Vergnolle et al., 2010; Thomas et al. , 2016; Wallace et al. , 2016).

Observations suggest that this occurs between the seismogenic zone and the frictionally stable zone (Fig. 1) surrounding the critical value of ordinary earthquakes nucleation; i.e., on parts of faults where the behavior is transitional between frictional properties on the rocks and slow, steady deformation (Scholz, 1998; Dragert et al., 2001; Beroza and Ide, 2009; Abe and Kato, 2014; Watkins et al., 2015), but some investigation reveal that have been observed in shallow region (Davis et al. , 2011; Nishimura , 2014; Saffer and Wallace , 2015; Valée et al. , 2015; Yamashita et al. , 2015; Wallace et al. , 2016). The Figure 1 shows two stability regions and transition zones: at shallow depth, and on the base of seismogenic layer (Scholz, 1998) which is the focus of this paper.





Scholz (1998) determined that at the border of the stability transition there is a region in which self-sustained oscillatory ($SSO$) motion occurs into the conditionally stable region, below the critical point, and slow earthquakes are nucleated. These oscillations are observed in presence or absence of external forces such as vibrations from neighbor faults but eventually tend to stabilize.

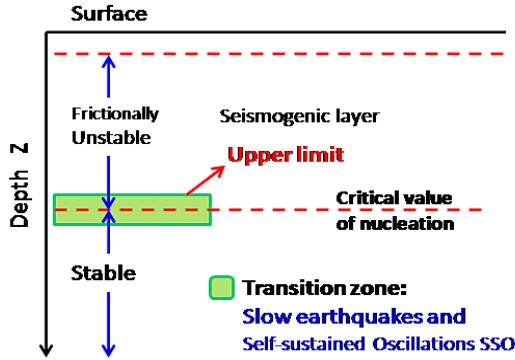

**Figure 1.** Transition zone related to slow earthquakes nucleation. The first doted red line indicates the lowest limit of the shallow frictionally stable region. The second one shows the deeper limit of the seismogenic layer (upper limit of the deeper frictionally stable region). Self-oscillatory behavior is observed in the second transition.

Slow earthquakes occur in a variety of stick-slip (Ide et al., 2007). Watkins et al. (2015) say that observational studies have provided information for their characterization: variability in duration, recurrence and propagation velocity, but the mechanism of slow earthquakes is still unclear. Experimental data show that the physical and mechanical parameters that control changes in slow slip events are the rates of convergence, frictional parameters and effective normal stress, under rate and state dependent constitutive properties of Dieterich-Ruina friction law (Dieterich, 1979; Ruina, 1983; Watkins et al., 2015; Marone et al., 2015;
Scuderi et al., 2016).

    The Dieterich-Ruina friction law has been successfully used to reproduce the stick slip behavior in the models of earthquakes dynamics and slow earthquakes. Spring-block models have reproduced these events when coupled with rate and state dependent friction laws, and contrary, models which have been used laws velocity-weakening friction and constant friction have not been successful (Abe and Kato, 2014). Beroza and Ide (2009) infer that simple friction laws by themselves do not provide an
explanation for the complex behavior of slow earthquakes. Some investigations have suggested that fluids play an important role in this mechanism (Brown et al., 2005; Ide et al., 2007; Wallace et al. , 2016). Some researches support this idea, conclusions from revised literature stablish that failures are lubricated on the shear area regardless of the composition of the rocks and the frictional weakening mechanism involved (Faulkner et al., 2010; Di Toro et al., 2011).

    An important issue of the rate and state dependent friction law is that it is totally macroscopic, i. e., it describes the frictional
properties of the system rather than microscopic mechanism which is responsible for the dissipation (Ruina, 1983; Carlson and Batista, 1996; Batista and Carlson, 1998). Experimental research on the role of fluids in the ordinary earthquakes mechanism



has captured specific features associated with the molecular layer lubrication on the border between two surfaces in contact, with different frictional properties when considering lubricants in volume and dry interfaces (Gee et al. , 1990; Yoshizawa and Israelachvili, 1993; Reiter et al., 1994; Carlson and Batista, 1996; Batista and Carlson, 1998; Amendola and Dragoni, 2013). The initiation slip in the microscopic scale is associated with a shear melting transition in the lubricant layer (Carlson and Batista, 1996) so that microscopic scales contribute to better understanding of friction mechanism.

A path to study slow earthquakes is through spring-block model for ordinary earthquakes, because both the slow and ordinary earthquakes are related by the critical value of nucleation. Some spring-block models display complex oscillatory behavior associated with the transition zone (Gu et al., 1984; Abe and Kato, 2013, 2014). Gu et al. (1984) and Abe and Kato (2013), used a spring-block model where complex oscillatory behavior was found near to a critical value of the earthquakes nucleation. This behavior is presented as changing when there is a variation in any parameter related to the critical value.

Batista and Carlson (1998) in a spring-block model introduced an alternative friction law, where the state variable is interpreted in terms of the shear melting of the lubricant (molecular layer of lubricant) between solid surfaces in contact. They considered that by incorporating the microscopic mechanism it could determine other behavior and found that the transition from steady sliding to stick slip is typically discontinuous and sometimes hysteretic. This transition is associated with a subcritical Hopf bifurcation (set of seismic parameters for the critical point of nucleation). Gu et al. (1984), Abe and Kato (2013), and Batista and Carlson (1998) observed a sudden and discontinuous onset in the amplitude of oscillations at the bifurcation point. Complex oscillatory behavior was observed in these cases, at surrounding of transition.

In terms of dynamical systems based on spring-block model, the presence of oscillations and self-oscillations can be explained by Hopf bifurcation mechanism (Avrutin et al., 2014; Kengne et al., 2014; Xin-YouMeng and Hai-Feng Huo, 2014; Wang et al., 2014). Some nonlinear dynamical systems show $SSO$ motion (Strogatz, 1994), one of them is relative to the earthquake physics mechanism (Scholz, 1998). The $SSO$ behavior would be explained in the context of the complex system of faults, such that the modeling of movement in a single failure would be affected by external forces (Chelidze et al. , 2005). These forces can be generated due to vibrations or stress transferred from neighboring faults, that make the system going from a limit cycle to another, leaving different paths of recurrence (Chelidze et al. , 2005; Dragoni and Santini, 2010; Dragoni and Piombo, 2011).

In this paper the main objective is to propose an upper limit for the $SSO$ region in the frictionally unstable zone (Fig. 1). The upper limit is a function of mathematical and numerical relations in terms of seismic and frictional parameters. This limit describes the complexity of the oscillatory movement in the nearest region to the critical point of nucleation. Another objective is determine which is the role of the fluids in this region. The oscillatory behavior of the system is studied through analysis of the Madariaga's spring-block model (Erickson et al., 2008) complemented by the Stribeck's effect (Alvarez-Ramírez et al., 1995; Andersson et al., 2007). The Stribeck's effect shows the transition from dry interfaces or lubricated at the border until separated by a layer of lubricant as a shear melting. This effect takes into account the microscopic mechanism between the contacting surfaces during displacement. The Dieterich-Ruina-Stribeck oscillator (Fig. 2) depicts the behavior of the kinetic mechanism between tectonic blocks in the earth's crust undergoing stick-slip effects from friction. The oscillator is coupled to





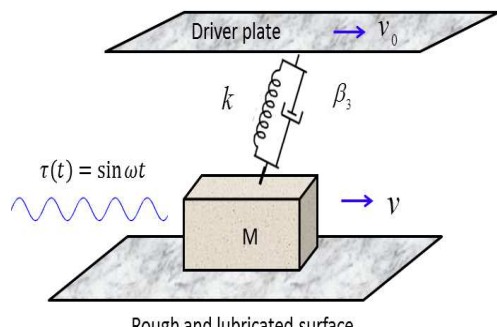

**Figure 2.** Dieterich-Ruina-Stribeck one degree of freedom oscillator. $v_0$ is a reference velocity, $v$ is de block velocity, $k$ is the constant of deformation, $\beta_3$ is the dynamical viscosity coefficient. $\tau(t)$ is an external and perturbing force with angular frequency $\omega$.

a driven plate by the rheological properties of rocks, and to a static plate by frictional properties. The slider is on the rough and lubricated surface. The relatives displacement, rate of displacement and acceleration are given by $u = x - v_0 t$, $\dot{u} = v - v_0$, and $\ddot{u} = \dot{v}$ respectively where $x$ is the block displacement and $v = \dot{x}$. An external periodic perturbation $\tau(t)$ is introduced in order to show that this system exhibits $SSO$ behavior between the critical point of nucleation and the region limited by

the proposed upper limit. In Section 2 the model is proposed, and the linearized system, stationary solution and criterion for frictional stability are analyzed; in Section 3 the oscillatory behavior is described, the upper limit of $SSO$ region is proposed and numerical simulation is provided, finally, in Section 4 a brief discussion of the outcomes and conclusions are presented.

## 2   Nonlinear Dynamical System

### 2.1   The model

The faults are lubricated in the shear area (Faulkner et al., 2010; Di Toro et al., 2011), as the most sliding contacts. A model based in a slider block (Fig. 2) is essentially a mechanic representation, and the frictional components are velocity function. Andersson et al. (2007) explain the Stribeck's effect as follows. The friction force varies with the sliding speed depending on the extent to which the interacting contact surfaces are running under boundary, mixed (fluid in the union between asperities), or full film lubrication (a layer of fluid as a shear melting). Even dry contacts show some behavior similar to that in lubricated

contacts in that they have a higher static friction than dynamic or sliding friction. In lubricated sliding contacts, the friction decreases with increasing sliding speed until a mixed or full film situation is obtained, after which the friction in the contact can be constant, increase, or decrease somewhat with increasing sliding speed due to viscous and thermal effects. This transition is the named Stribeck's effect and has been formulated as (Jacobson, 2003; Andersson et al., 2007):

$$F_s(v) = \beta_1 sign(v - v_0) + \beta_2 e^{-\mu(v)} sign(v - v_0) + \beta_3 v, \tag{1}$$





where $\beta_1$ is the Coulomb friction; $\beta_2 = F_{max} - \beta_1$ (with $F_{max}$ as the upper limit of static force); $\beta_3$ represents the dynamical viscosity coefficient of the fluid; and $\mu$ denotes a slip constant or decay parameter for the mixed lubrication. Note that the slider block velocity is relative to velocity of driver plate. The $sign(v - v_0)$ is negative when $v < v_0$, zero as $v = v_0$ and positive otherwise. We dismissed the first term of Eq.(1) because it is regarded in the Dieterich-Ruina friction law. The $sign(v - v_0)$ indicates the direction of movement and it is removed, letting the Dieterich-Ruina friction law indicates the direction, but regard the transition from dry surfaces to mixed lubrication (second term of Eq.(1)), and finally to a layer of fluid between the surfaces as a shear melting. The equation is as follows:

$$F_0(v) = \beta_2 e^{-\mu(v)} + \beta_3 v, \tag{2}$$

on the other hand the well known phenomenological friction law of Dieterich-Ruina is introduced in rocks mechanics to capture experimental observations of steady state and transient friction that depends on the displacement history effects (state variable $\theta$) and velocity (Dieterich, 1979; Ruina, 1983; Dieterich and Kilgore, 1994), one of its formulations is given by two equations Ruina (1983):

$$F_{dr}(v, \theta) = \theta + A \ln(v/v_0), \qquad \dot{\theta} = -(v/L)[\theta + B \ln(v/v_0)]. \tag{3}$$

First equation represents the frictional stress under stable state ($\dot{\theta} = 0$), and the second one corresponds to evolution of state variable $\theta$ that evolves with time, slip, and normal stress history (Dieterich and Kilgore, 1994). $L$ is a characteristic sliding distance over which $\theta$ evolves (required distance to renew the contact population). $A$ and $B$ are material properties, and we assume $B > A > 0$ for frictional instability.

Now it is possible to derive from Eq.(2) and (3) an alternative friction law taking account the macroscopic and microscopic mechanism (frictional properties and dissipation). The equation of motion derived from Eq.(2) and (3) gives a third order system differential equations, a similar expression is given in Erickson et al. (2008), but we complemented their system by the term of Stribeck's effect $F_0(v)$:

$$
\begin{aligned}
\dot{\theta} &= -(v/L)[\theta + B \ln(v/v_0)], \\
\dot{u} &= v - v_0, \\
\dot{v} &= -(1/M)[ku + F_{dr}(v, \theta)] - (1/M)F_0(v) + \tau(t).
\end{aligned}
\tag{4}
$$

Defining new variables $\hat{\theta}$, $\hat{v}$, $\hat{u}$ and $\hat{t}$ (Erickson et al., 2008): $\theta/A = \hat{\theta}$, $v/v_0 = \hat{v}$, $u/L = \hat{u}$, $t(v_0/L) = \hat{t}$, the dimensionless system is given by the equations

$$
\begin{aligned}
\dot{\hat{\theta}} &= -\hat{v}[\hat{\theta} + (1 + \varepsilon)\ln(\hat{v})], \\
\dot{\hat{u}} &= \hat{v} - 1, \\
\dot{\hat{v}} &= -\gamma^2[\hat{u} + (1/\xi)(\hat{\theta} + \ln(\hat{v}))] - \alpha F_0(\hat{v}) + \tau(\hat{t}).
\end{aligned}
\tag{5}
$$

where $\alpha = \{\alpha_2, \alpha_3\} = \frac{L}{v_0^2 M}\{\beta_2, \beta_3\}$. The frictional parameters $\alpha$ are associated with frictional coefficients from the Stribeck's effect, and

$$\alpha F_0(\hat{v}) = \alpha_2 e^{-\hat{\mu}\hat{v}} + \alpha_3 \hat{v}, \qquad \hat{\mu} = v_0 \mu. \tag{6}$$



$x = (\hat{\theta}, \hat{u}, \hat{v})$ is the dimensionless state variable, $\hat{\theta}$ stands for the measurement of contact with asperities from the Dieterich-Ruina friction law; $\hat{u}$ is the dimensionless relative displacement between the block and the upper plate and $\hat{v} > 0$ is the dimensionless velocity of the block.

Parameters $\Pi = (\varepsilon, \xi, \gamma)$ are given as follows: $\varepsilon = (A - B)/A$; $\xi = kL/A$ and $\gamma = \sqrt{(k/M)}(L/v_0)$, associated with stress drop during displacement, deformation and the oscillation frequency, respectively. The equation (5) is referred to the system of Dieterich-Ruina-Stribeck (DR-S) where $\tau(\hat{\mathbf{t}}) = [0, 0, \tau(\hat{t})]^T$ is the periodical and deterministic external force $\tau(t) = \sin \omega t$, here $\omega$ is the angular frequency. We named unperturbed system when $\tau(t) = 0$ and perturbed system otherwise. We will denote $(\hat{\theta}, \hat{u}, \hat{v}) := (\theta, u, v)$, $\hat{t} := t$, and $\hat{\mu} := \mu$.

## 2.2 Stationary solution at equilibrium point

The stationary solution $x = x^\star$ of the system (5) has the components

$$x^\star = (\theta^\star, u^\star, v^\star) = (0, \eta, 1), \qquad u^\star = \eta = (\alpha_2 e^{-\mu} + \alpha_3)/\gamma^2, \qquad \gamma > 0, \tag{7}$$

$\eta$ corresponds to the relative position of the single slider block. At $x^\star$ the plate and the block have the same velocity and the measure of the asperities contact is zero. Note that $\theta^\star$ and $v^\star$ do not depend on $\Pi$ but $u^\star$ depends on frequency oscillation $\gamma$ (consequently on $kL$) and frictional constants of Stribeck's effect, both are associated with the energy dissipation.

The local and asymptotical stability of $x^\star$ is analyzed with the indirect method of Lyapunov that consists in the analysis of the eigenvalues of the Jacobian matrix from the linearized system of Eq.(5) around the stationary solution (Khalil, 1996). Let $x^\star$ locally asymptotically stable mean that every solution of the system $(\theta, u, v)$, starting near of the stationary solution, it remains at the surrounding of $x^\star$ all the time, and eventually the solution converges to $x^\star$ (convergence to frictional stability). Let us denote the Jacobian matrix as $D_f(x^\star) = (\partial f_i(x)/\partial x_j) \mid_{x^\star}$, for $i, j = 1, 2, 3$, where $f(x)$ is the vectorial field or right side of (5) with $\tau(t) = 0$; and let $\lambda_i$ be the eigenvalues of $D_f(x^\star)$:

$$D_f(x^\star) = \begin{pmatrix} -1 & 1 & -(1+\varepsilon) \\ 0 & 0 & 1 \\ -\gamma^2/\xi & -\gamma^2 & -\gamma^2/\xi - \phi, \end{pmatrix} \tag{8}$$

where $\phi = \alpha_2 \mu e^{-\mu} - \alpha_3$. The polynomial characteristic of (8) is given by:

$$P(\lambda) = a_0 \lambda^3 + a_1 \lambda^2 + a_2 \lambda + a_3, \tag{9}$$

whose coefficients are in terms of seismic parameters $\Pi$, and frictional coefficients $\alpha$ and $\mu$

$$a_0 = 1 \qquad a_1 = 1 + \gamma^2/\xi + \phi \qquad a_2 = \gamma^2(1 - \varepsilon/\xi) + \phi \qquad a_3 = \gamma^2. \tag{10}$$

The dynamical system of earthquakes is a naturally dissipative phenomena due to this feature the dissipativity condition of the stationary solution is required. Thus locally the system is dissipative at $x^\star$ if $a_1 = -\text{Trace} D_f(x^\star) = \sum_{i=1}^{3} \lambda_i < 0$ that is



true under the condition:

$$\frac{Mv_0^2}{L} + A + \beta_2 \mu e^{-\mu} > \beta_3; \tag{11}$$

Equation (11) comes directly from $a_1 < 0$ and the values of $\gamma$, $\varepsilon$, $\xi$, and $\phi$. The equation (11) is the necessary condition for the system to be sub-damped , and oscillations can be observed; moreover due to $a_3 = -\det D_f(x^\star) < 0$, $x^\star$ is hyperbolic. Through the analysis of the eigenvalues of $D_f(x^\star)$ we will explain what implies a hyperbolic equilibrium point related to oscillatory behavior.

## 3 Oscillatory Behavior

The earthquakes dynamics is a nonlinear oscillatory phenomenon (Gu et al., 1984; De Sousa Vieira, 1995; Levin, 1996; Chelidze et al. , 2005; Maloney and Robbins, 2007; Erickson et al., 2008; Dragoni and Santini, 2010; ; Amendola and Dragoni, 2013; Abe and Kato, 2014); where the nonlinear complex behavior is attributable to the friction forces. The analysis of the oscillatory behavior is explored in this section.

### 3.1 Analysis of Eigenvalues

The equilibrium point $x^\star$ is locally asymptotically stable if the real part of all the eigenvalues is negative, i.e. $Re(\lambda_i) < 0$, and it is unstable if $Re(\lambda_i) \geq 0$ for one or more eigenvalues of $D_f(x^\star)$. In order to determine how many eigenvalues of $D_f(x^\star)$ are real or conjugate complex we use the Descarte's rule of signs to analyze the roots of polynomial characteristic. Under the condition (11), signs of coefficients (10) are

$$(+, +, sign(a_2), +). \tag{12}$$

If $sign(a_2) > 0$ there are two possibilities: all eigenvalues are negative real, i. e., $Re\{\lambda_i\} < 0, Im\{\lambda_i\} = 0$ or one eigenvalue is negative real and the other two are complex conjugates; the last statement corresponds to the oscillatory behavior. From the Eq.(10) and the fact of $sign(a_2) > 0$ we deduce

$$\varepsilon < \xi\psi \qquad \psi = 1 + \phi/\gamma^2, \tag{13}$$

as a necessary condition for stability but it is not sufficient. In Section 3 is given a sufficient condition to stability.

On the other hand, if $sign(a_2) < 0$, there are possibly two positive real eigenvalues, i.e., $Re\{\lambda_k\} > 0, Im\{\lambda_k\} = 0$ and one negative real; for this case there are no conjugate complex eigenvalues, hence non-oscillatory behavior is observed. For all cases there is one negative real eigenvalue, and the other two could be complex conjugates, or positive real. Figure 3 shows the locus of the real part of two eigenvalues corresponding to the oscillatory and the non-oscillatory behavior, it describes the relationship between parameters $\Pi$. The graph for the negative real eigenvalue was omitted because we focused on complex eigenvalues.



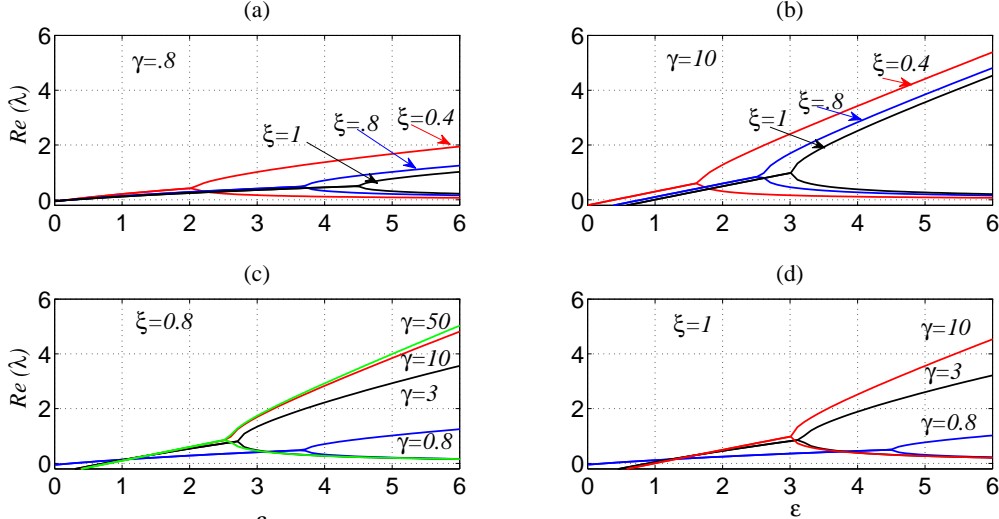

**Figure 3.** Locus of eigenvalues for different values of $\xi$ and $\gamma$. The Figure shows the real part of eigenvalues as a function of parameters $\varepsilon$, $\xi$ and $\gamma$. For (a) and (b) fixed $\gamma = 0.8, 10$, respectively; (c) and (d) shows eigenvalues for fixed $\xi = 0.8, 1$ respectively.

The oscillatory behavior is located before the branching, after which the system ceases to oscillate. The Locus of the eigenvalues for different values of $\xi = 0.4, 0.8, 1$ are shown in Fig. 3(a), (b) for $\gamma = 0.8$ and $\gamma = 10$, respectively. The range of $\varepsilon$ decreases with the increasing of $\gamma$ and the decreasing of $\xi$. Fig. 3(c) and 3(d) describe the same behavior as explained above, and it was observed that for values of $\gamma > 10$ the range of $\varepsilon$ for oscillatory behavior is almost equal, although after the branching point with the increasing of $\gamma$, one of the eigenvalues increases rapidly, i. e., $kL$ increases making biggest the system stiffness or $L$ increases; and the other one tends to zero, more quickly.

We are interested in the type of hyperbolic stationary solution $x^\star = (0, \eta, 1)$: it has a stable manifold; i.e., $Re\{\lambda_i\} < 0, Im\{\lambda_i\} = 0$, and a unstable manifold that generates oscillations in a plane, i.e., $Re\{\lambda_k\} > 0, Im\{\lambda_k\} \neq 0$ (Campos-Cantón et al., 2010) because it was found a set of parameters that satisfies the necessary condition (13) for stability, within an unstable and oscillatory region which is around of the Hopf bifurcation (set of seismic parameters that satisfies the critical value of nucleation; Fig. 1). The region with these features is proposed for the region of self-sustained oscillations ($SSO$) and consequently for the slow earthquakes zone. The $SSO$ region is in the unstable region and will be explored in the follow sections.

## 3.2 A Hopf Bifurcation, Oscillatory Range ($OR$) and the Self-sustained Oscillations Region ($SSO$)

The presence of oscillations in physical systems can be explained through the mechanism of Hopf bifurcation. When three eigenvalues exist two of them are complex conjugates and the other is a non-zero real, a Hopf bifurcation occurs (Fig. 4) if the real part of the complex eigenvalues cross the imaginary axis. Periodic orbits and limit cycles are either created or destroyed





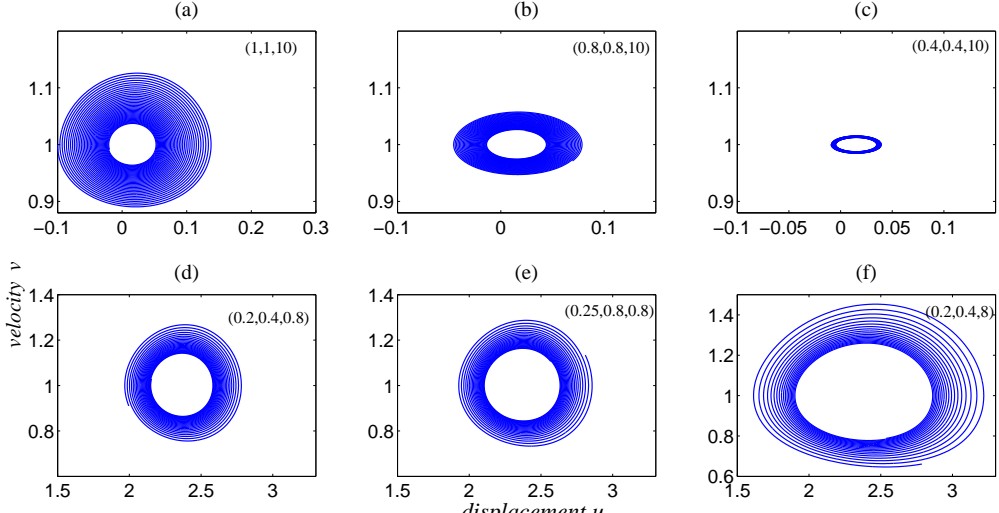

**Figure 4.** Unperturbed system $\tau(t) = 0$. Projection of attractor onto the plane $(u, v)$. (a), (b) and (c) have the position of equilibrium point $u^\star = \eta$ near zero for $\gamma = 10$, $\xi$ decreases as it is the range of $u$. (d), (e) and (f), for $\gamma = 0.8$ have both $u^\star$ and the range of values for $u$ higher than (a)-(c).

for the nearest values of the Hopf Bifurcation (Guckenheimer and Holmes, 1983). If all neighboring trajectories approach the limit cycle then it can be said that the limit cycle is stable or an attractor. Stable limit cycles are important because they model systems that exhibit $SSO$ behavior (Strogatz, 1994), therefore starting at Hopf Bifurcation, the oscillatory behavior around it will be analyzed.

There are three necessary conditions in order to a Hopf bifurcation may occurs: (i) The existence of an equilibrium point, $f(x^\star) = 0$ (ii) the jacobian matrix $D_f(x^\star)$ has a couple of eigenvalues on the imaginary axis, i. e., $Re\{\lambda_{1,2}\} = 0$ y $Im\{\lambda_{1,2}\} \neq 0$; and (iii) the cross velocity of eigenvalues through imaginary axis must be different to zero.

For any system with three variables the conditions (ii) and (iii) for obtaining a Hopf bifurcation are given in terms of the

10 polynomial characteristic coefficients $a_1$, $a_2$, and $a_3$ from Eq.(10)(Baca , 2007). The cross velocity is given by the derivative $r'(\Pi_0) = \frac{d}{d\gamma}(Re\{\lambda(\Pi)\})|_{\gamma=\gamma_0} \neq 0$, where $\Pi_0 = (\varepsilon_0, \xi_0, \gamma_0)$ is a set of fixed values and $\lambda(\Pi)$ is an eigenvalue of $D_f(x^\star)$ at $\Pi$. The derivative is with respect to the bifurcation parameter, $\gamma$ (oscillation frequency). The cross velocity is

$$r'(\Pi_0) = \frac{a_3'(\Pi_0) - b^2 a_1'(\Pi_0) + \lambda_0 a_2'(\Pi_0)}{2(\lambda_0^2 + b^2)}, \qquad a_1(\Pi_0) = -\lambda_0 \text{ and } a_2(\Pi_0) = b^2. \tag{14}$$

The polynomial from Eq.(9) has a couple of imaginary roots $\lambda_{1,2}$ if there is a $\Pi$ such that the following two relations are

15 satisfied

$$a_3 = a_1 a_2, \qquad a_2 > 0. \tag{15}$$



If $\Pi = \Pi_0$ satisfies the Eq.(15), then the complex roots of Eq.(9) with real part zero are determined by

$$\lambda_{1,2} = \pm i b_0 \qquad b_0 = \sqrt{a_2}, \qquad \lambda_3 = -\frac{a_3}{a_2}. \tag{16}$$

From Eq.(10) and (16), the eigenvalues of (8) for the Hopf bifurcation are given by

$$\lambda_{1,2}(\Pi_0) = \pm i \sqrt{\gamma_0^2 - \frac{\gamma_0^2 \varepsilon_{HB}}{\xi_0} + \phi}, \qquad \lambda_3(\Pi_0) = -\frac{\gamma_0^2}{\gamma_0^2 - \frac{\gamma_0^2 \varepsilon_{HB}}{\xi_0} + \phi}. \tag{17}$$

The Jacobian matrix (8) has two conjugate complex eigenvalues with real part positive for values of $\varepsilon$ which depend on the fixed values $\gamma_0$ and $\xi_0$. These eigenvalues correspond to oscillatory region in the unstable regime, and they are observed from the Hopf bifurcation to the beginning of bifurcation as it is depicted in the Fig. 3, this interval is called the oscillatory range $OR$. We want to find the limits of $OR$ and the $SSO$ in terms of seismic parameters.

From Eq.(10) and (15) it is determined $\varepsilon_{HB}$, i. e., the $\varepsilon$ value for the Hopf bifurcation

$$\varepsilon_{HB} = \xi_0 \left[ 1 - \frac{1}{1 + \frac{\gamma_0^2}{\xi_0} + \phi} + \frac{\phi}{\gamma_0^2} \right]. \tag{18}$$

$\varepsilon_{HB}$ is the lower limit of $OR$. The upper limit is determined by the discriminant of the third order polynomial $a_0 \lambda^3 + a_1 \lambda^2 + a_2 \lambda + a_3$:

$$D = \left( \frac{3a_2 - a_1^2}{9} \right)^3 + \left( \frac{9a_1 a_2 - 27a_3 - 2a_1^3}{54} \right)^2, \tag{19}$$

If $D > 0$ there are two complex conjugate roots and one is real, if $D = 0$ all are real roots, and at least two are equal; and if $D < 0$ all roots are real and unequal. We are interested in $D = 0$, which implies that the oscillatory behavior finishes (there are not complex roots). From Eq.(9), (10), and (19)

$$D = \left[ \frac{3 \left( \gamma_0^2 - \frac{\gamma_0^2 \varepsilon_{D=0}}{\xi_0} + \phi \right) - \left( 1 + \frac{\gamma_0^2}{\xi_0} + \phi \right)^2}{9} \right]^3 +$$

$$\left[ \frac{9 \left( 1 + \frac{\gamma_0^2}{\xi_0} + \phi \right) \left( \gamma_0^2 - \frac{\gamma_0^2 \varepsilon_{D=0}}{\xi_0} + \phi \right) - 27\gamma_0^2 - 2 \left( 1 + \frac{\gamma_0^2}{\xi_0} + \phi \right)^3}{54} \right]^2 = 0,$$

and we can resolve by $\varepsilon_{D=0}$, according to Fig. 3. The $OR$ for $\varepsilon$ is in the interval

$$OR \in (\varepsilon_{HB}, \varepsilon_{D=0}). \tag{20}$$

In the $OR$, the necessary condition (13) for stability is satisfied by a set of parameters $\Pi$ around the Hopf bifurcation; the region with these features is proposed for the $SSO$. The proposed interval for $SSO$ is

$$SSO \in (\varepsilon_{HB}, \xi_0 \psi_0), \qquad \psi_0 = 1 + \frac{\phi}{\gamma_0^2}. \tag{21}$$



| $\gamma$ | $\xi$ | $\varepsilon_{HB}$ | $\pm(\lambda_{1,2})_{HB}$ | $\varepsilon_{D=0}$ | $SSO \in (\varepsilon_{HB}, \xi_0\psi_0)$ | $r'_\varepsilon(\Pi_0)$ |
|---|---|---|---|---|---|---|
| 0.8 | 0.4 | 0.1981 | 0.5030i | 2.081779482130000 | (0.1981,0.3562) | 0.3042 |
| | 0.8 | 0.2499 | 0.6083i | 3.704570372000000 | (0.2499,0.7123) | 0.2058 |
| | 1.0 | 0.2534 | 0.6385i | 4.507300000000000 | (0.2534,0.8904) | 0.1749 |
| 10 | 0.4 | 0.3981 | 0.6313i | 1.668570861615000 | (0.3981,0.3997) | 0.4981 |
| | 0.8 | 0.7931 | 0.8911i | 2.601243068598171 | (0.7931,0.7994) | 0.4963 |
| | 1.0 | 0.9894 | 0.9954i | 3.018387963893687 | (0.9894,0.9993) | 0.4953 |

**Table 1.** Relationship between parameters $\Pi$ on the oscillatory range for Figures 3 (a) and (b). $\varepsilon_{HB}$: value of the Hopf Bifurcation; $(\pm\lambda_{1,2})_{HB}$: eigenvalues on the imaginary axis; $\varepsilon_{D=0}$: value when the discriminant of the polynomial characteristic is equal to zero, i. e. upper limit of $OR$; $SSO$ is the interval of self-sustained oscillations, and $r'(\Pi_0)$ is the cross velocity of eigenvalues.

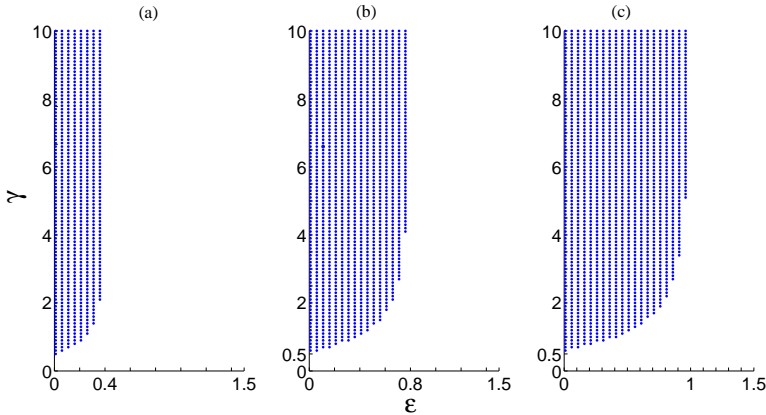

**Figure 5.** Stability region for homogeneous system ($\tau(t) = 0$), as a function of $(\varepsilon, \xi, \gamma)$ for fixed values of $\xi$. (a) $\xi = 0.4$, (b) $\xi = 0.8$, and (c) $\xi = 1$. The necessary and sufficient conditions are satisfied.

Some numerical results are summarized in Table 1. A sufficient condition for local and asymptotical stability comes from Eq.(18)

$$\varepsilon < \varepsilon_{HB}. \tag{22}$$

Equation (22) is equivalent to the Routh-Hurwitz criterion. For the region with this sufficient condition, all eigenvalues of the jacobian matrix (8) have negative real part and the equilibrium point $x^\star = (0, \eta, 1)$ is a sink (Perko et al., 2001).

The relationship between the parameters $\Pi$ associated with the necessary condition (13) and sufficient condition (22) are described in Fig. 5 (a), (b) and (c), for fixed $\xi = 0.4, 0.8, 1$, respectively. By means of numerical simulation the stability was computed and found for values of $\gamma > 0.5$.



Our interest is for the case that $\varepsilon = (B - A)/A > 0$, or $B - A > 0$; which means that the stress drop is negative and consequently an unstable regime is observed; under this assumption, $\xi\psi$ it is a positive amount implying that $\psi > 0$. Equation (13) is a necessary condition for stability, which is maintained within a set of values of parameters where the system is unstable (Table 1).

In the DR-S model any small perturbation in the system can change the dynamical behavior. If the DR-S system is subject to perturbations from neighboring faults, the seismic fault enters in a limit cycle, but it does not remain long there due to intervening stress perturbations (Dragoni and Santini, 2010).

### 3.3   The System Under Forcing Conditions

We are interested in the oscillatory behavior when the values of parameters $\Pi$ are nearest to the Hopf bifurcation. We numeri-
cally explored the effects of an external, deterministic and periodic force $\tau(t)$ acting on the system (5), for $\Pi=\{0.25,0.8,0.8\}$, $\alpha=\{0.2,0.1\}$, and $\mu=3$. Such effects are illustrated by varying the angular frequency $\omega$ from $\tau(t) = \sin\omega t$. This numerical analysis is helpful visualizing patterns in the dynamic of the system, especially those related to oscillatory behavior, such as limit cycles and periodic orbits. Figure 6 shows numerical results of typical oscillations projected onto the plane $(u, v)$ and

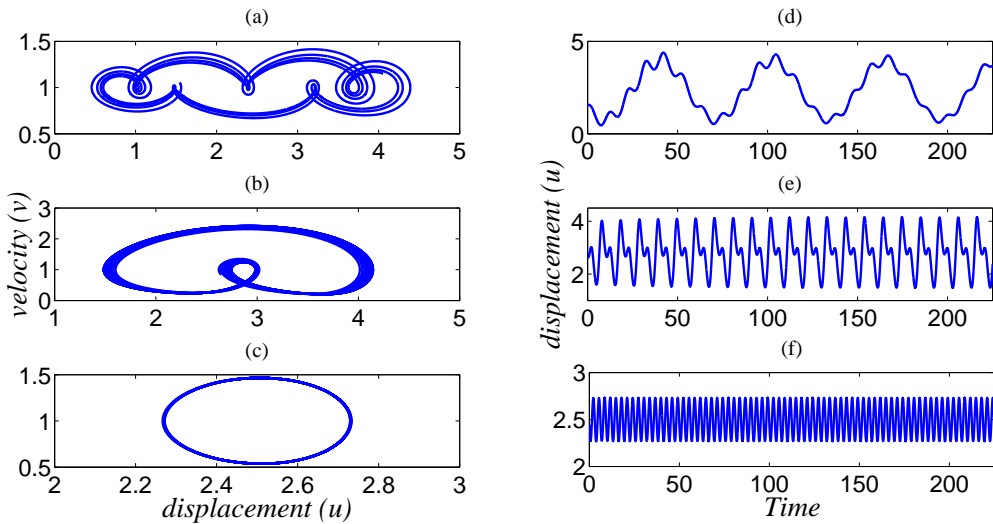

**Figure 6.** Projection of the attractor onto the plane $(u, v)$ and time series of the perturbed system (5). (a), (b) and (c) show the projection onto the plane for $\omega = 0.1, 1.2, 2$, respectively. (d), (e) and (f) display their respective time series for $u$.

time series generated by Eq.(5). The system (5) presents different behaviors when the parameter $\omega$ takes values in the interval
[0.1,2]. For example when $\omega \in [0.1, 0.8]$, a type of complex behavior for low frequencies is observed (Fig. 6 (a)). Figure 6 (b) shows that for values of $\omega \in [0.8, 1.3]$ periodic orbits of period two are found. For angular frequencies $\omega > 1.3$ the flow of the system converges to a limit cycle as shown in Fig. 6 (c). On the other hand, Fig. 6(d) to (f) describe the time series of $u$ when




$\omega$ is 0.1, 1.2 and 2, respectively. The motion is periodical, this behavior emphasizes the periodic motion of the DR-S. The time series for low frequencies ($\omega < 0.8$) are more complex than the other cases.

### 3.3.1 Bifurcation Diagram for Unperturbed System

Qualitative changes in the dynamic of the system are better understood through bifurcation analysis such that when a control

parameter is varied the bifurcations show the transitions or instabilities of the system. The unperturbed system is considered when $\tau(t) = \sin \omega t = 0$, i. e., $\omega = 0$. In order to show that the behavior of the unperturbed system displays SSO, the control parameter $\gamma = \sqrt{(k/M)}(L/v_0)$ is varied, which in turn is related to frequency of oscillation of the slider block and to the characteristic longitude of displacement $L$. Numerical results are for fixed $\xi = 0.8$, and the value for $\varepsilon = 0.25$ that holds the necessary condition (13).

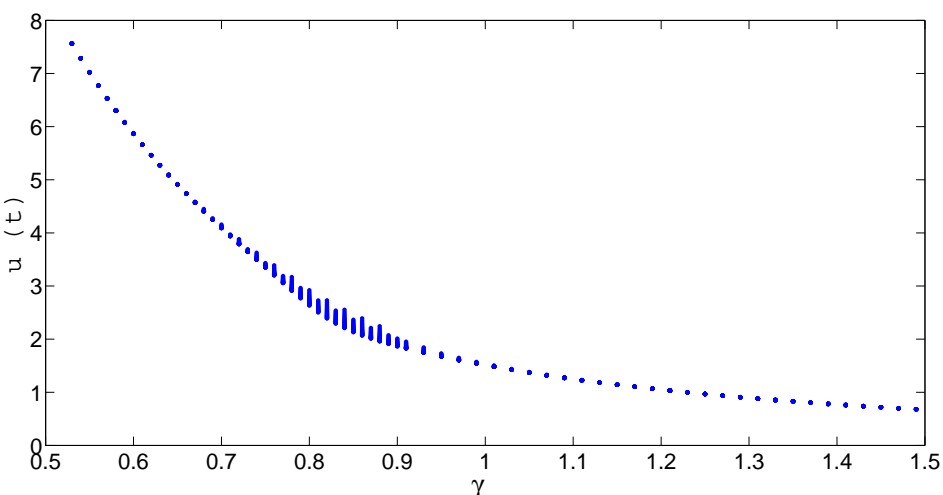

**Figure 7.** Bifurcation diagram for unperturbed system $\tau(t) = 0$: Bifurcation parameter $\gamma$ versus local maximum of time series $u_{lmax}(t)$, $\Pi = \{0.25, 0.8, \gamma\}$.

Under the necessary condition (13), the system without external perturbations oscillates and multi-periodic orbits are created over an approximate range of values $\gamma \in [0.6, 1]$. Eventually, the flow of the system converges to a limit cycle as is depicted in Fig. 7 and 8. This oscillatory behavior is observed when $\Pi$ are nearest to the Hopf bifurcation in absence of external forces. The range of $u_{lmax}(t)$ decreases when $\gamma$ increases.

### 3.3.2 Bifurcation Diagram for Perturbed System

According to previous outcomes, we considered the forced system $\tau(t) = \sin \omega t$ centered at $\omega = \{0.1, 1.2\}$, for fixed $\xi = 0.8$, and two values of $\varepsilon$. Two values for $\varepsilon = \{0.25, 1\}$ are explored. The first value holds the necessary condition and the second one does not. The bifurcation parameter is $\gamma$. We named case one to the analysis of bifurcation with fixed $\varepsilon = 0.25$, and second case





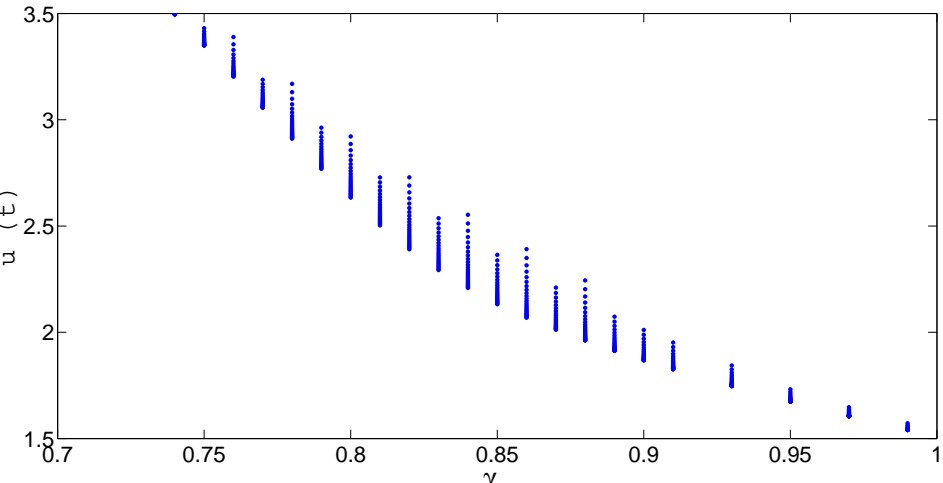

**Figure 8.** Zoom of Figure 7.

for $\varepsilon = 1$; for both cases $\xi = 0.8$ is fixed . The bifurcation diagram for case one is displayed in Fig. 9, it shows the qualitative

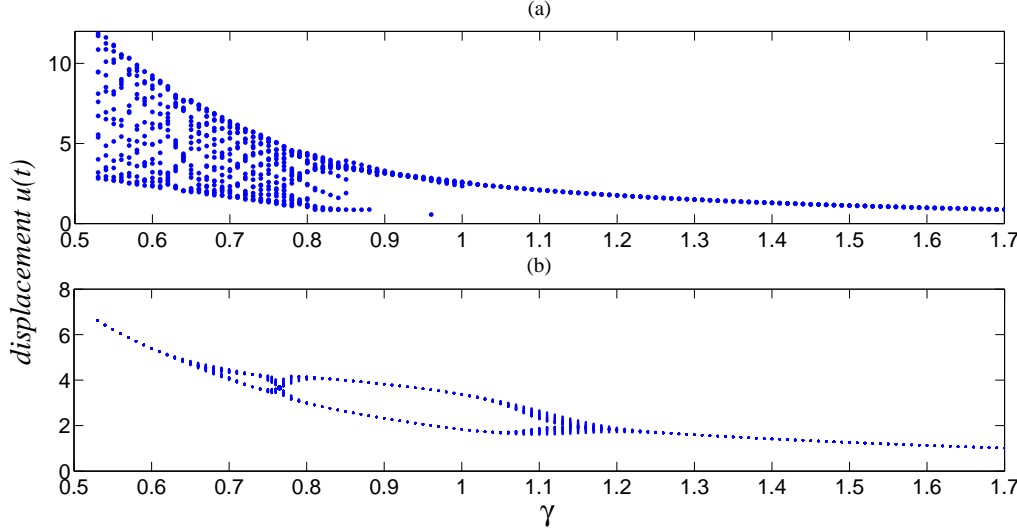

**Figure 9.** Bifurcation diagram $\gamma$ versus local maximum from the time series $u_{lmax}(t)$. $\Pi = \{0.25, 0.8, \gamma\}$, $\mu = 3$, $\alpha = \{0.2, 0.1\}$, sub-damped system. (a) $\omega = 0.1$, and (b) $\omega = 1.2$.

behavior of system relative to the variable of position $u$ and oscillation frequency $\gamma$. The figure displays $\gamma$ versus its local




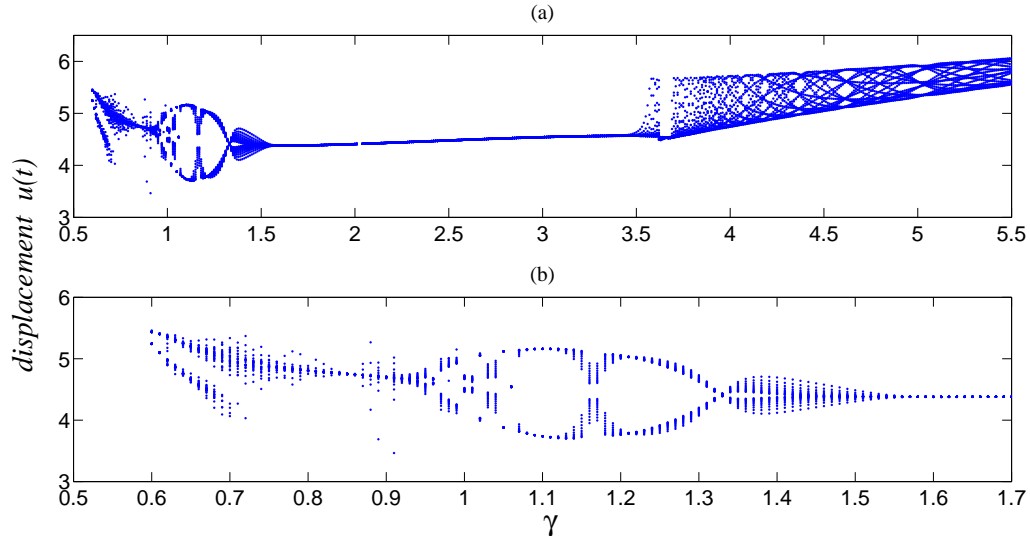

**Figure 10.** Bifurcation diagram $\gamma$ versus local maximum from time series $u_{lmax}(t)$, $\omega = 1.2$, $\Pi = \{1, 0.8, \gamma\}$ $\mu = 3$, $\alpha = \{0.2, 0.1\}$, sub-damped system. (a) There are three types of behavior: Type I, II and III. (b) Zoom of Type I.

maximum of time series $u_{lmax}(t)$. Figure 9 (a) for $\omega = 0.1$ shows that the most orbits are weakly attractive until approximate values of $\gamma > 0.9$ when limit cycles are observed.

On the other hand, Fig. 9 (b) for $\omega = 1.2$ shows limit cycles, which are observed approximately when $\gamma \in [0.55, 0.66)$, $\gamma > 1.17$, and $\gamma = .765$; whereas there are two points of bifurcation from orbits of period one to orbits of period two at

5 $\gamma = 0.66$ and $\gamma = 0.765$. The orbits of period two have the form of Fig. 6 (b), and the limit cycles take the form as in Fig. 6 (c). The orbits are strongly attractive, which suggests that they could be stable at least during a short period. According to the necessary condition for stability. The behavior of $u_{lmax}(t)$ relative to $\gamma$ decreases for both $\omega = 0.1$ and $\omega = 1.2$ because after $\gamma$ reaching the values 0.9 and 1.17, respectively, the system falls into the limit cycle, in such a way that when $\gamma$ increases, the range of values for $u_{lmax}(t)$ decrease. The type of behavior displayed in Fig. 9 is expected when $\varepsilon < \xi\psi$.

The region of $SSO$ could be numerically explained by this analysis; if the system is perturbed slightly by external forces $\tau(t)$, it always returns to the standard cycle.

The bifurcation diagram for case two is displayed in the Fig. 10 (a) for $\omega = 1.2$. The necessary condition (13) fails; and the dynamic of the system shows three types of behavior: behavior Type I (Fig. 10 (b)) is observed approximately at $\gamma \in (0.6, 1.55)$. Type I shows periodic orbits of period one and two that appear to be alternating, bifurcations from orbits of period one to period

two occur, then the system reaches a limit cycle at $\gamma = 1.55$ but now with Type II behavior for approximate values of $\gamma \in (1.55, 3.5)$. In the Type II $u_{lmax}(t)$ it increases maintaining the limit cycle until behavior Type III is observed approximately for $\gamma \in (3.5, 5.5)$, which displays periodic orbits with different period. The behavior displayed in Fig. 10 corresponds to unstable and oscillatory region outside of the $SSO$.



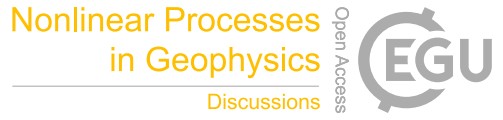

## 4 Discussion and Conclusions

We have analyzed the DR-S model which describes the kinetic mechanism during an earthquake. The system displays richness in their oscillatory dynamic behavior: from attracting cycles of one and two periods to a limit cycle and multi-periodic orbits, depending on the parameters values in the necessary condition for stability (13). The necessary condition is maintained even

for a set of parameters within the frictionally unstable region. This behavior was studied by bifurcation diagrams through the Hopf bifurcation mechanism.

The complex oscillatory behavior discussed in this paper is determined by the variation of parameter $\gamma$, fixed values $\varepsilon$ and $\xi$, and the necessary condition for stability (13). The necessary condition depends on the parameter $\gamma$, and this depends on the characteristic length $L$, suggesting that the complex oscillatory behavior should be observed for a range of values for the

parameter $\gamma$. The unperturbed and perturbed cases have shown how the system describes behavior as it is found in systems with $SSO$, this is in the case when the necessary condition (13) is maintained (Fig. 7 and 9). There are self-oscillations that create multi-periodic orbits, but eventually converge to a limit cycle and the range of $u_{lmax}(t)$ decreases when $\gamma$ increases.

The $\gamma$ parameter seems to be relevant, and consequently the characteristic longitude $L$. Lapusta et al. (2000), Lapusta and Rice (2003), and Abe and Kato (2013) derived some conclusions concerned to $L$ from laboratory experiments and numerical

simulations. They associated $L$ with the size of small earthquakes, with frequency, and $SSO$. If $L$ decreases then the block is frictionally less stable, i. e., there are more frequent displacements; moreover they found under a range of parameters values the oscillations can change from periodic to aperiodic and conversely, but the border of the transition was not defined.

Under the assumption $B > A > 0 \Rightarrow \varepsilon > 0$ and Eq.(18), Hopf bifurcation occurs for any $\gamma \geq 0.5$ within the standard $0.4 <$ $\xi_0 < 1$, moreover, $\varepsilon_{HB} \to \xi_0$ when $\gamma$ increases in such way that for any $\gamma \geq 10$ the values of $\varepsilon_{HB}$ are equal. For these values $\varepsilon$

is bounded in $0 < \varepsilon_{HB} < 1$. This would have some implications related to the slow earthquakes nucleation. For $SSO$ region, approximately for $0.5 < \gamma \leq 10$ the oscillation frequency and fluid are determinant for the system have unstable oscillations. For small values of $\gamma$ there are more transitions in the dynamical oscillatory behavior, when the necessary condition (13) holds and does not it as it is shown in Fig. 7, 9, and 10. By other hand for $\gamma \geq 10$ there are less transitions, neither the fluid and frictional coefficients of Stribeck in the medium affect to the oscillatory behavior ($\phi/\gamma^2 \to 0 \Rightarrow \psi \to 1$). This general behavior

for small $\gamma$ seems to be independent on the angular frequency, $\omega$, of the external force because even the unperturbed system displays this behavior. The function of the $\omega$ variation is the generation of the dominant type of orbits (one, two, or more periods) and its frequency of transition which is bigger when the necessary condition fails. The complex oscillatory behavior is dependent of small $\gamma$ values, and of the necessary condition for stability (13) that involves fluids from Stribeck's effect.

The relevance of $\gamma$ seems to be associated with the presence of fluid. Erickson et al. (2008) determined that the stationary

solution $x^\star = (0, 0, 1)$ is independent of seismic parameters, the relative position of the block and the plate will not depend on the frictional coefficients or seismic parameters; the system would be under the limit of the stable/unstable regime where the big earthquakes are nucleated. At this solution the block and the plate have the same velocity and there is no contact with asperities, hence there is no displacement between the plate and the slider block.





Complementarily we found if Stribeck's effect is added to the Madariaga's Model then the stationary solution $x^\star = (0, \eta, 1)$ has the displacement or relative position as a function of seismic parameter $\gamma$. The parameter $\gamma$ is associated with the oscillation frequency of the block as well as the frictional constants related to fluid through $\psi$ in the necessary condition (13). The relative position $u$ is not necessarily zero although the relative displacement is. The role of frequency oscillations and fluid is decisive

for dissipation and consequently it is also for the block position. If $\gamma$ is large enough then $x^\star \to (0, 0, 1)$, i. e., if $\gamma$ or $kL$ increases, then $u^\star \to 0$. The last statement could be interpreted as $k$ increases making the system stiffer or $L$ increases making slower the stress drop under some conditions.

The stability analysis of $x^\star$ was through eigenvalues of Eq.(8). If the real part is negative for all eigenvalues then the stationary solution is a sink and the system is in the stable regime, the medium properties break any nucleation or propagation of earthquakes. Scholz (1998) determined a critical value for frictional stability/instability under elastic medium, with an

oscillator coupled to Dieterich friction law:

$$\overline{\sigma}_c = \frac{-kL}{a - b} \tag{23}$$

that depends on the rocks properties, point nucleation and frictional parameters $a - b$, $k$, y $L$. The critical value $\overline{\sigma}_c$ corresponds to the normal effective stress. When any $\overline{\sigma}$ holds $\overline{\sigma} \geq \overline{\sigma}_c$, then there are changes in the frictional properties, such changes

unchained earthquakes. This phenomena is named frictional instability. He reported that the $SSO$ are into the stable regime, below the critical value of nucleation (23) (Fig. 1), on complementary way this investigation reveals that the critical value (23) of Scholz (1998) have got to the upper limit of $SSO$ in the unstable region, and it is related to the upper limit of $SSO$ in the frictionally unstable region regardless Stribeck's effect.

The relation between the $SSO$ (21) and (23) comes from the definition of $\Pi = (\varepsilon, \xi, \gamma)$ and the statement $(A - B) = \sigma(a - b)$

(Ruina, 1983) as follows. From Eq.(23) $\overline{\sigma}_c(b - a) = kL \Rightarrow (B - A) = kL \Rightarrow (B - A)/A = kL/A \Rightarrow \varepsilon = \xi$, by other hand $\varepsilon = \xi\psi \Rightarrow \overline{\sigma}_c(b - a)/A < kL\psi/A \Rightarrow \overline{\sigma}_c = -kL\psi/(a - b)$, a corrected value for $SSO$ is as follows

$$1 < \overline{\sigma}_c^* \equiv \frac{-kL\psi}{(a - b)} \equiv \overline{\sigma}_c\psi, \tag{24}$$

which combines frictional parameters of Dieterich-Ruina friction law and Stribeck's effect. Equations (13) and (24) are equivalent. If $\gamma > 10$, $\psi \to 1$ and $\overline{\sigma}_c^* \to \overline{\sigma}_c$.

The upper limit of $SSO$ behavior is a function of seismic parameters and frictional coefficients concerned to fluids although this was stablished for the base of seismogenic layer, it is likely probably it could be applied to the shallow transition zone in addition with the parameters and constants related to the slip-hardening (Ikari et al., 2013). The fluid presence involves the frequency of oscillation of the block as a very important element to dissipation and consequently with the stationary solution (equilibrium point of the system) as well as in the upper limit proposed for slow earthquakes zone. The characteristic length $L$

has a primary relevance on the results of this research.

*Competing interests.*   The authors declare that they have no conflict of interests.



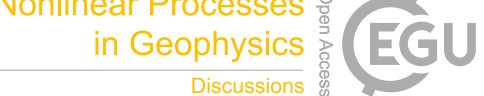

*Acknowledgements.* This study was supported by CONACyT (support 44731), the Departments of Applied Mathematics and Applied Geo-sciences of Instituto Potosino de Investigación Científica y Tecnológica (IPICYT), and the Instituto de Geología de la Universidad Autónoma de San luis Potosí, in México. Special thanks to Raquel Jaramillo for reviewing spelling and grammar.



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
