# Peer review of "An upper limit for slow earthquakes zone: Self-oscillatory behavior through the Hopf bifurcation mechanism from a model of spring-block under lubricated surfaces"

_Nonlinear Processes in Geophysics, 2016_

## Referee Comment (RC1) · Anonymous Referee #1 · 3 Dec 2016

The paper is a contribution to fault mechanics based on the theory of dynamical systems. The authors consider an ideal seismogenic region made of a shallow seismogenic layer and an underlying stable zone where fault creep takes place. The aim is to determine the depth of the border between the two regions. It is suggested that the border zone is the origin of slow earthquakes, a phenomenon that has not yet received a satisfactory explanation. The problem is studied by considering a slider block model with a rate and state-dependent friction law: the conditions under which self-sustained oscillatory motion occurs are investigated. Overall, the paper is well written and the developments are clearly exposed. There is a good review of the state of art. The

authors derive interesting results with implications as to slow earthquake nucleation and the role of fluids in the border region. I recommend publication after a moderate revision according to the following remarks.

My chief objection to the paper is that too little space is devoted to the link between the model and real seismogenic regions. The authors should establish a neat correspondence between the values of the model parameters and real conditions in the Earth. They should provide at least one example, assigning specific values to the parameters and deriving their consequences in terms of dimensional quantities, such as the depth of the border, the thickness of the border zone, the fluid content, the frequency of the perturbation, and so on. This would make the paper more appealing to a wider audience.

Secondly, the authors should check definitions and dimensions of the quantities involved in the model. It seems that nondimensional quantities are introduced starting from equation (5). If equation (1) has dimensions, is Fs a force per unit length? According to (3), the quantities A, B and theta have the same dimensions. The variable x is defined as the block displacement at page 4, line 3, but the same symbol is used for the dimensionless state vector at page 6, line 1.

Some minor corrections are: Page 1, line 2: Ruinas's should be Ruina's. Page 4, line 2: "relatives" should be "relative". Page 4, line 11: "velocity function" should be "velocity dependent". Page 5, lines 4-7: the sentence is not clear and should be rephrased. Page 7, line 15: Descarte's should be Descartes'. Page 16, line 13: "longitude" should be "length". Page 17, line 26: "stablished" should be "established". Figure 1, caption: "doted" should be "dashed". Figure 2, caption: "de" should be "the".

---

## Author Comment (AC1) · 4 Jan 2017

We thank to Anonymous Referee number 1 for his comments on the article. We believe that his suggestions and recommendations are very timely and very supportive to enrich this document. We take his observations into account and they were attended as follows:

**RFC1 (Referee's comments)**: My chief objection to the paper is that too little space is devoted to the link between the model and real seismogenic regions. The authors should establish a neat correspondence between the values of the model parameters

and real conditions in the Earth. They should provide at least one example, assigning specific values to the parameters and deriving their consequences in terms of dimensional quantities, such as the depth of the border, the thickness of the border zone, the fluid content, the frequency of the perturbation, and so on. This would make the paper more appealing to a wider audience.

**AR (author's replay)**: Our main contribution is more related to the proposals of a formal pattern in the study of SSEs, and a first approximation of the limit of the transition zone in terms of seismic and frictional parameters. We agree that the real example would make the paper more appealing to a wider audience, however the parameters considered for slow earthquakes are still being studied experimentally and by means of simulations, but there is still not something precise, so giving a real example is complicated.

However, an approximation of some of the parameters considered in this article are discussed in Watkins et al. 2015 (See references there in). Their study seek to model observations with a simulator to reproduced reported characteristics of SSEs in Cascadia through variations in (A-B), normal stress, and convergence rate. The estimates of these parameters do not adjust to the SSEs in Cascadia with respect to the recurrence intervals. Fluids and variations in the width transition zone might affect the recurrence times, among other factors.

Estimates of these frictional parameters are given below (Watkins et al., 2015): The slip amounts of SSEs are of order of cm but the average slip amount of smaller events are unknown. The efective normal stress in the range of 3-9 MPa produce fault slip consistent with some observed SSEs., $B - A$ is in the range (0.0015 to 0.003) of the slow slip section. At the top of the slow slip section 0.003 and 0.001 at the base, $A \approx 0.02$, $L$ is in the range 1-50 $\mu$m (real $L$ is unknown), the Rate of convergence (10 a 50 mm/year ) represents the range of convergence rates of subduction zones where SSEs are observed with GPS. The critical value $K_c = (A - B)\sigma/L$ depend on $L$. These

parameters could vary depending on the region that SEEs occur.

By other hand, viscosity=0.1 (nondimensional) has been used in earthquake models (Carlson et. Al., Reviews of Modern physics, Vol. 66, No.2, 1994), but the estimation of the real viscosity depends on the region.

**RFC1**: Secondly, the authors should check definitions and dimensions of the quantities involved in the model. It seems that nondimensional quantities are introduced starting from equation (5). If equation (1) has dimensions, is $F_s$ a force per unit length? According to (3), the quantities A, B and theta have the same dimensions? The variable $x$ is defined as the block displacement at page 4, line 3, but the same symbol is used for the dimensionless state vector at page 6, line 1.

**AR**: From the equation (1), page 4, the units of $F_s$ are Newtons. The first term is the Coulomb friction (dry or lubricated at the border), the second term (friction effect for mixed lubrication) is the diference between the maximum static force and the Coulomb friction; and the last term is the viscous friction, all the forces have units of Newtons (see Andersson et. al. 2007).

The complete form of the equation (3) is given by

$$F_{dr} = \sigma[\theta + A\ln(v/v_0)], \qquad d\theta/dt = -(v/L)[\theta + B\ln(v/v_0)]. \qquad (1)$$

$F_{dr}$ is the frictional stress, $\sigma$ is the normal stress (constant) and $[\theta + A\ln(v/v_0)]$ is the coefficient of fault friction which is nondimensional, i.e., $F_{dr}/\sigma = [\theta + A\ln(v/v_0)]$.

Derived from experimental and mathematical approaches, A and B are constants (intrinsec properties of material), " $\theta$ is a weighted average of $-\ln(v/v_0)$ over the distance $L$"( Ruina, 1983, page 10363, paragraph 3 ).
The variable $x$ assigned for the displacement of the block was replaced by the variable $y$ (page 4, line 3).

**RFC1**: Some minor corrections are: Page 1, line 2: Ruinas's should be Ruina's. Page 4, line 2: relatives should be relative. Page 4, line 11: velocity function should be velocity dependent. Page 5, lines 4-7: the sentence is not clear and should be rephrased. Page 7, line 15: Descarte's should be Descartes'. Page 16, line 13: longitude should be length. Page 17, line 26: stablished should be established. Figure 1, caption: doted should be dashed. Figure 2, caption: de should be the.

**AR**: We did the corrections of the misspelling words in the following pages: page 1, line 2; page 4, line2; page 4, line 11; page 7, line 15; page 16, line 13; page 17, line 26; Figure 1, caption; and Figure 2, caption.

Page 5, lines 4-7. We take into account the comment on the need to explain or rewrite this part, in addition we relocate equation (2) and part of the explanation after equation (4), page 5.

From the second equation of the system (4) on page 5, we infer that the block will oscillate with respect to the position of the plate. This equation tells us whether the oscillator is to the right or to the left with respect to the driver plate ($v$ is the oscillator displacement rate and $v_0$ that of the plate). Although the direction of the displacement of the system is always forward; If $v - v_0 > 0$, i.e., if $v > v_0$, eventually the oscillator will be more advanced than the plate; On the other hand if $v - v_0 < 0$, i.e., $v < v_0$ eventually the oscillator will be to the left of the plate. One of the objectives when introducing the function $sign(v - v_0)$ is indicate this effect; assigning value 1, for the first case, -1 for the second case, and zero if $v = v_0$. We eliminate the sign function because this effect is already considered in the second equation of the system (4).

---

## Referee Comment (RC2) · Anonymous Referee #1 · 10 Jan 2017

I take note of the authors' reply that the paper is a preliminary study that needs further refinements in order to be applied to real seismogenic regions. I suggest that this is explicitly stated in the conclusions, possibly indicating which are the following steps that should be made toward application. The considerations that the authors make in the reply with reference to Watkins et al. (2015) indicate a possible link with observations and should be included in the paper. The dimensional problem in some equations has been solved. Misspellings have been corrected and the sentence at page 5 has been rephrased and expanded. With the above proviso, I confirm my positive opinion on the paper.

---

## Author Comment (AC2) · 12 Jan 2017

We appreciate all the comments and suggestions from the Anonymous Referee Num.1. These have been very supportive to improve the document. In response to the last comment we mentioned the following.

**RC1 (Referee's comments)**: I take note of the authors' reply that the paper is a preliminary study that needs further refinements in order to be applied to real seismogenic regions. I suggest that this is explicitly stated in the conclusions, possibly indicating which are the following steps that should be made toward application. The considerations that the authors make in the reply with reference to Watkins et al. (2015) indicate a possible link with observations and should be included in the paper. The dimensional problem in some equations has been solved. Misspellings have been corrected and the sentence at page 5 has been rephrased and expanded. With the above proviso, I confirm my positive opinion on the paper.

**AR (authors' replay)**: On page 2, lines 5-10, we mention, in a summary and general way, the information that is available from observational and experimental studies for the characterization of SSEs, also the parameters involved in these investigations are mentioned (Watkins et al., 2015; Marone et al., 2015; Scuderi et al., 2016).

We agree that the following should be mentioned at the end of the conclusions (pages 17, 18):
Although this investigation is more related to the proposal of a formal pattern in the study of SSEs, and with a first approximation of the upper limit of the transition zone, this is considered as a preliminary study in order to be applied to the real seismogenic regions. However, the parameters considered for slow earthquakes are still being studied experimentally and by means of simulations, and there is still not something precise.

The study of SSEs in Cascadia (Watkins et. al. 2015) indicates a possible link between the observational and experimental data with the parameters involved in the most of models of earthquake's physic coupled to the Dieterich-Ruina's friction law. The slip amounts of SSEs are of order of cm but the average slip amount of smaller events are unknown. The effective normal stress in the range of 3-9 MPa produce fault slip consistent with some observed SSEs., $B - A$ is in the range (0.0015 to 0.003) of the slow slip section. At the top of the slow slip section $B - A$ is 0.003 and 0.001 at the base, $A \approx 0.02$, $L$ is in the range 1-50 $\mu$m (real $L$ is unknown), the rate of convergence (10 a 50 mm/year ) represents the range of convergence rates of subduction zones where SSEs are observed with GPS. These parameters could vary depending on the

region that SEEs occur. On the other hand, the critical value $K_c = (A - B)\sigma/L$ depend on $L$; viscosity=0.1 (nondimensional) has been used in earthquake models (Carlson et. al., 1994), but the estimation of the real viscosity depends on the region.

The proposed upper limit for the SSEs zone includes the fluids and oscillation frequency (and consequently, $L$), through $\psi$. They might be introduced into the simulations and experiments in order to see which are the implications over the recurrence times, duration and velocity of SSEs in real seismogenic regions. A final step would be using scaling laws for SSEs for real parameters that are not included nether experimental nor simulation data, such as the stiffness $K_c$ and viscosity, take into account the specific characteristics of the fault.

Furthermore, the reference of Carlson et al. (1994) is added to the bibliography.

---

## Referee Comment (RC3) · Anonymous Referee #2 · 28 Mar 2017

This paper addresses the problem in geophysics of finding the depth of the border between a thin seismogenic layer and an underlying stable layer. This is important because the border area is thought to be where slow earthquakes occur. Determination of the region size must be based on a knowledge of the mechanism for such events. Hence a correct estimation of the border depth holds out the possibility of predicting such earthquakes.

I take as my starting point the idea that the geophysics community welcomes work of an accomplished theoretical standard, that good theory is to be valued and nurtured (that is not clear in other areas, such as parts of engineering, where design considerations can drown out detailed theory).

So I can only be disappointed in this paper. It labours a standard analysis, misses out both key points and important results and then leaves the reader bereft of a valid geophysical conclusion. Also the exposition is poor (concepts not explained well) and the English is, in places, dire.

Where to begin? Well, the 3 page introduction nods in the direction of many different papers both in geophysics and in theory. I am not an expert in the former, but the ground appears well covered, giving a sense of why the area is important and why simple Coulomb friction will not do, for sound geophysical reasons. But the theory references are less convincing. For example, I would never refer to Avrutin et al for the Hopf bifurcation.

Section 2 introduces equations (5) which are the main subject of the paper. Again, the justification for the system seems fine. But then the problems begin. It took me a while to work out that section 2.2 is about the unforced (not *unperturbed*) equations. But it is the treatment of the characteristic polynomial (not *polynomial characteristic*) that grates. In situations like this, you have to immediately state and use the Routh Hurwitz (RH) criteria. These are the industry standard used to determine the necessary and sufficient conditions needed to ensure that the equilibrium of the unforced equations is stable. You do not get that from this paper (true, RH is mentioned later on p11, but only in passing).

Section 3 begins badly. I accept that earthquakes are nonlinear and that this nonlinearity comes about because of friction. But the eigenvalues correspond to the linear problem and only tell us about the equilibrium solution. We need to work a lot harder to understand the role of nonlinearity.

What should really happen is that the RH criteria should be front and centre of the paper. These give you the clear limits on parameters that guarantee a stable equilibrium. You have 3 eigenvalues, so for stability you need the real part of all 3 to be negative. If that happens, your unforced quake dies away exponentially - no oscillations. Then if you are happy with damped linear oscillations (are these meant to be the slow earthquakes?), you need two of the eigenvalues to be complex conjugate (they have to be conjugate as your characteristic polynomial has real coefficients). But again the real parts of all 3 eigenvalues have to be negative. Then as you vary parameters, you want to avoid the real parts of the complex conjugate pair crossing the imaginary axis (otherwise you get a Hopf bifurcation).

All of this is in the paper, but so hard to find and interpret. What is needed is a clearer structure, starting with RH and then some good bifurcation diagrams.

Now you have your Hopf bifurcation and you get oscillations in your unforced system. But now the question is: is it a subcritical or supercritical Hopf bifurcation? The latter is

not very interesting. The former is very dangerous, where a small perturbation before the Hopf is reached can lead to either decaying oscillations or a jump to a large sustained oscillations in the system. Which do you have in this paper? It seems to me to be of extreme importance to know which it is, since the monitoring of a crucial parameter by geophysicists would be doomed to failure if it were a subcritial Hopf (a sustained quake would be triggered way before you reach what you think is your danger point).

Section 3.3 reintroduces the forcing term and consists of some numerical simulations. But here a big opportunity is missed. There is a wealth of theoretical work done on forced systems near Hopf bifurcations (going back many years), whose results show clearly that everything depends on what type of Hopf bifurcation you have in the first place. I found this section to be very poor. One paper with some good references on this topic is Yanyan Zhang and Martin Golubitsky *Periodically Forced Hopf Bifurcation* SIAM J. Applied Dynamical Systems 10 (4) 1272–1306 (2011).

The last section is a discussion of the results, devoid of any real connection with geophysics. Perhaps that is too much to ask. But I feel it is not too much to ask that straightforward theory be applied correctly.

Finally, the writing is poor. Ideas from dynamical systems theory are mangled and confused in a way that I would need an hour to unpick. Same with the English: too many examples to deal with. Let me just mention the second sentence of the abstract: "The mathematical springblock model is generated by considering the Dieterich-Ruinas's friction law and the Stribeck's effect." would work much better if it were something like "The mathematical spring-block model includes Dieterich-Ruina's friction law and Stribeck's effect" or "The mathematical spring-block model includes the Dieterich-Ruina friction law and the Stribeck effect." As it stands, the sentence incorrectly conflates two nouns, uses an awkward construction, uses the wrong possessive and misspells a name (Ruina, not Ruinas).

---

## Author Comment (AC3) · 17 May 2017

We appreciate the comments and suggestions from the Anonymous Referee number 2. We thank for his comments on the article. These suggestions and recommendations are very timely and very supportive to enrich this document. We take his observations into account and they were attended as follows:

**Referee's comment**. Where to begin? Well, the 3 page introduction nods in the direction of many different papers both in geophysics and in theory. I am not an expert in the former, but the ground appears well covered, giving a sense of why the area is

important and why simple Coulomb friction will not do, for sound geophysical reasons. But the theory references are less convincing. For example, I would never refer to Avrutin et al for the Hopf bifurcation.

**Author's replay**. We take the suggestion into account.

**Author's changes in manuscript**:
The reference from Avrutin et al. (2014) was removed. Section 3.3 and the references section include the following quotation:
Yanyan Zhang and Martin Golubitsky Periodically Forced Hopf Bifurcation SIAM J. Applied Dynamical Systems, 10 (4), 1272 - 1306 (2011).

**Referee's comment**. Section 2 introduces equations (5) which are the main subject of the paper. Again, the justification for the system seems fine. But then the problems begin. It took me a while to work out that section 2.2 is about the unforced (not unperturbed) equations. But it is the treatment of the characteristic polynomial (not polynomial characteristic) that grates. In situations like this, you have to immediately state and use the Routh Hurwitz (RH) criteria. These are the industry standard used to determine the necessary and sufficient conditions needed to ensure that the equilibrium of the unforced equations is stable. You do not get that from this paper (true, RH is mentioned later on p11, but only in passing).

**Author's replay**. The reason we do not address much about the sufficient condition for stability is because we are more interested in the necessary condition (13), which is fulfilled for values close to the Hopf bifurcation in the unstable regime. Specifically we are interested in oscillatory behavior, so we analyze the form of eigenvalues to focus our study on complex conjugate eigenvalues with positive real part.
We agree to restructure section 3.1. The Routh Hurwitz criteria must go in this section.
**Author's changes in manuscript**:
In Section 2.2, line 11, page 6, we added: with tau (t) = 0. This indicates that the analysis is about unforced equations.

Lines 7 and 8 on page 8 are relocated on line 14 of page 7; Followed by lines 1-8 on page 11 and fig 5. Lines 20-22 of page 7 are inserted and rephrased into one of the conditions for the Routh Hurwitz criteria.

**Referee's comment**. Section 3 begins badly. I accept that earthquakes are nonlinear and that this nonlinearity comes about because of friction. But the eigenvalues correspond to the linear problem and only tell us about the equilibrium solution. We need to work a lot harder to understand the role of nonlinearity.

**Author's replay**. We agree. We need to work a lot harder to understand the role of nonlinearity in many of the geophysical phenomena.
In the spring-block model, the logarithmic term in the Dieterich-Ruina friction law has introduced greater difficulty in solving the problem. Due to the nonlinear term, analytic integration has not been possible, and even numerical solutions present challenges because of the logarithmic term (Erickson *et al.*, 2008). The linearized system analysis is very useful to describe some features of the nonlinear system about steady state solution (Gu *et al.*, 1984; Shkoller and Minster, 1997; Erickson *et al.*, 2008).

It is important to say that we started our nonlinear analysis as in Gu *et al*. (1984), Shkoller and Minster (1997), and Erickson *et al*. (2008) determining the equilibrium point. This equilibrium point is of great interest since this is where the steady sliding occurs. This solution is very important in the analysis of the dynamic system that characterizes the earthquakes mechanism. The linearized system allows us to simplify our analysis. The linearized system gives a qualitatively correct image of the phase portrait near $x^\star$, since it is a sink or saddle point in Perko's definition (2001), so for the original nonlinear system, $x^\star$ it is really a sink or a saddle Point (Strogatz, 1994, Page 151, see the references therein).

We are interested in the saddle point which is a hyperbolic point in the sense that the Jacobian matrix has at least one eigenvalue with negative real part and at least one

eigenvalue with positive real part. The steady-state solution is a hyperbolic point for parameter values in the oscillatory interval (OR) (Eq. (20) page 10).

**Author's changes in manuscript**:
We added a clarifying text in section 2.2 after line 15 on page 6: the first paragraph of this replay.
The reference Shkoller and Minster (1997) was added in the references section.
In the introduction of section 3 line 11, page 7 we added:
We use the full nonlinear term in the numerics in sections 3.2 and 3.3.

**Referee's comment**. What should really happen is that the RH criteria should be front and centre of the paper. These give you the clear limits on parameters that guarantee a stable equilibrium. You have 3 eigenvalues, so for stability you need the real part of all 3 to be negative. If that happens, your unforced quake dies away exponentially - no oscillations. Then if you are happy with damped linear oscillations (are these meant to be the slow earthquakes?), you need two of the eigenvalues to be complex conjugate (they have to be conjugate as your characteristic polynomial has real coefficients). But again the real parts of all 3 eigenvalues have to be negative. Then as you vary parameters, you want to avoid the real parts of the complex conjugate pair crossing the imaginary axis (otherwise you get a Hopf bifurcation). All of this is in the paper, but so hard to find and interpret. What is needed is a clearer structure, starting with RH and then some good bifurcation diagrams.

**Author's replay**. Our analysis does not focus on the stable regime, but on the unstable (Page 7, lines 25-28, page 8, lines 7-11, and Figure 3; page 10, lines 20-24; page 12, lines 1-3). The earthquakes can only be generated in the unstable regime. Our interest is in the unstable regime near the critical nucleation value (near the hopf bifurcation, where parameters approximate values for the critical value of nucleation). Specifically, we analyze the oscillatory region: one of the eigenvalues is negative real and two are

complex conjugate with positive real part.

The mechanism of slow earthquakes is not very clear yet, but one of the theories is that they are generated in the transition zone, between the frictionally unstable and stable region, with parameters values near the critical value of nucleation, hence the importance of starting from the hopf bifurcation and analyzing the oscillatory behavior in their environment, particularly in the unstable regime. In previous studies a complex behavior has been observed in this neighborhood. The oscillatory behavior is very sensitive to variations in the values of the parameters; on the other hand Scholz (1998) deduces that there are self-oscillations in that region, therefore we start from the hypothesis that in the region of slow earthquakes there are damped oscillations depending on the values of the parameters, from this statement we derive the proposal of the upper limit of the slow earthquakes zone (page 2, lines 1-4, page 3, lines 6-25, Page 2, Figure 1).

**Referee's comment**. Now you have your Hopf bifurcation and you get oscillations in your unforced system. But now the question is: is it a subcritical or supercritical Hopf bifurcation? The latter is not very interesting. The former is very dangerous, where a small perturbation before the Hopf is reached can lead to either decaying oscillations or a jump to a large sustained oscillations in the system. Which do you have in this paper? It seems to me to be of extreme importance to know which it is, since the monitoring of a crucial parameter by geophysicists would be doomed to failure if it were a subcritial Hopf (a sustained quake would be triggered way before you reach what you think is your danger point).

**Author's replay**. From the Hopf bifurcation we varied the bifurcation parameter $\gamma$ in such a way that the parameter values be either before and after crossing the Hopf bifurcation plane. In terms of the flow in phase space, a supercritical Hopf bifurcation occurs when a stable spiral changes into a unstable spiral surrounded by nearly eliptical limit
Interactive comment

cycle. From the Hopf bifurcations shown in the Figure 4 (page 9) all simulations display behavior as in supercritical bifurcation. For example, in the figure attached as a supplement, starting at the Hopf bifurcation $\Pi = (0.25, 0.8, 0.8)$, Figures (a) and (e) show the parameter values have no crossed the Hopf bifurcation. Small disturbances decay after ringing for a while and stable spiral is observed. The block and the driver plate are moving at constant rate $v = 1$, and the relative position is $\eta$. This occurs when $\gamma > \gamma_{HB}$. On the other hand for $\gamma < \gamma_{HB}$, Figures (b) and (f) show the parameter values have crossed the Hopf bifurcation. The equilibrium state lose stability and unstable spiral is observed. This type of bifurcation is expected for smooth, non-catastrophic changes. The slow earthquakes are almost imperceptible because the displacement rate is very low compared to ordinary earthquakes and they are generated for parameter values around the critical value of nucleation. This argument and the numerical simulations leads us to infer that the Hopf bifurcation is supercritical within the proposed limits for unforced system. To find chaotic behavior or strange attractors with the non-forced system it is necessary to vary epsilon very far (Erickson, *et al.*, 2008) from the value of the Hopf bifurcation that we are analyzing.

However, Kostić *et al.* (2013)(Srdan Kostić, Igor Franović, and Kristina Todorović: Friction memory effect in complex dynamics of earthquake model, Nonlinear Dyn. (2013) 73:1933–1943 DOI 10.1007/s11071-013-0914-8) found chaotic behavior for small values of $\Pi$ by introducing delay time in the friction term. They found the two types of Hopf bifurcation depending on the variation of the delay time. Similarly, by introducing the external force $\tau(t)$ a subcritical Hopf Bifurcation could be given for some critical $\tau(t)$ and slight variation of the $\varepsilon$ and $\xi$ parameters. Disturbances do not allow the system to remain at an equilibrium point resulting in continuous oscillations or chaos. In the figure of the supplement (c) and (g) $\gamma > \gamma_{HB}$ and the values of $v$ and $u$ remain close to 1 and $\eta$ on average, whereas in the figures (d) and (h) the range for $v$ and $u$ is wider and variable for the case where $\Pi$ crosses the Hopf bifurcation. Continuous oscillations are found in both displacement and velocity only by varying the bifurcation parameter. For the analysis of the bifurcation type, for the system (5), the main challenge is the

numerical stiffness, due to the nonlinear logarithmic term. Determining critical value of $\Pi$ and $\tau(t)$ requires a more concrete study. We limit ourselves to the cases presented in the article, leaving it as opportunity area to explore the system.

**Author's changes in manuscript**:
The comments about this answer and the graph mentioned will be introduced in the new section 3.3.3 Numerical features from the Hopf bifurcation analysis.

**Referee's comment**. Section 3.3 reintroduces the forcing term and consists of some numerical simulations. But here a big opportunity is missed. There is a wealth of theoretical work done on forced systems near Hopf bifurcations (going back many years), whose results show clearly that everything depends on what type of Hopf bifurcation you have in the first place. I found this section to be very poor. One paper with some good references on this topic is Yanyan Zhang and Martin Golubitsky Periodically Forced Hopf Bifurcation SIAM J. Applied Dynamical Systems 10 (4) 1272–1306 (2011).

**Author's replay**. We added a brief introduction in section 3.3, page 12.

**Author's changes in manuscript**. Introduction in section 3.3:
This section aims to numerically describe the oscillatory behavior within and outside the range proposed for the SSO region (Eq. (21), page 10), under forcing and non-forced conditions. We want to prove numerically that the proposed upper limit determines the changes in oscillatory behavior, below and above this. For more theoretical background into the theory of periodically forced systems near a point of Hopf bifurcation, see Zhang And Golubitsky (2011) and references therein.

**Referee's comment**. The last section is a discussion of the results, devoid of any real connection with geophysics. Perhaps that is too much to ask. But I feel it is not too

much to ask that straightforward theory be applied correctly.

**Author's replay**. On page 2, lines 5-10, we mention, in a summary and general way, the information that is available from observational and experimental studies for the characterization of SSEs, also the parameters involved in these investigations are mentioned (Watkins *et al.*, 2015 ; Marone *et al.*, 2015; Scuderi *et al.*, 2016).

We agree that the real example would make the paper more appealing to a wider audience, however the parameters considered for slow earthquakes are still being studied experimentally and by means of simulations, but there is still not something precise, and even in simulations it is difficult to work with realistic values due to the logarithmic term, which makes the system sensitive to values of v close to zero, so giving a real example is complicated. The conclusions are given in terms of the implications of the parameters involved, according to previous results. However, an approximation of some of the parameters considered in this article are discussed in Watkins *et al.* (2015). Their study seek to model observations with a simulator to reproduced reported characteristics of SSEs in Cascadia through variations in (A-B), normal stress, and convergence rate. The estimates of these parameters do not adjust to the SSEs in Cascadia with respect to the recurrence intervals. Fluids and variations in the width transition zone might affect the recurrence times, among other factors.

**Author's changes in manuscript**:
We think that the following should be mentioned at the end of the conclusions (pages 17, 18):

Although this investigation is more related to the proposal of a formal pattern in the study of SSEs, and with a first approximation of the upper limit of the transition zone, this is considered as a preliminary study in order to be applied to the real seismogenic regions. However, the parameters considered for slow earthquakes are still being studied through observations, experiments, and by means of simulations, but there is still

not something precise.

The study of SSEs in Cascadia (Watkins, *et al.* 2015) indicates a possible link between the observational and experimental data with the parameters involved in the most of models of earthquake's physic coupled to the Dieterich-Ruina's friction law. The slip amount of SSEs are of order of cm but the average slip amount of smaller events are unknown. The effective normal stress in the range of 3-9 MPa produce fault slip consistent with some observed SSEs, $B - A$ is in the range (0.0015 to 0.003) of the slow slip section. At the top of the slow slip section $B - A$ is 0.003 and 0.001 at the base, $A \approx 0.02$, $L$ is in the range 1-50 $\mu$m (real $L$ is unknown), the rate of convergence (10 a 50 mm/year ) represents the range of convergence rates of subduction zones where SSEs are observed with GPS. These parameters could vary depending on the region that SEEs occur. On the other hand, the critical value $K_c = (A - B)\sigma/L$ depend on $L$; viscosity=0.1 (nondimensional) has been used in earthquake models (Carlson *et al.*,1994), but the estimation of the real viscosity depends on the region.

The proposed upper limit for the SSEs zone includes the fluids and oscillation frequency (and consequently, $L$), through $\psi$. They might be introduced into the simulations and experiments in order to see which are the implications over the recurrence times, duration and velocity of SSEs in real seismogenic regions. A final step would be using scaling laws for SSEs to determine the real values of parameters included either experimental and/or simulation data, such as the stiffness $K_c$ and viscosity, take into account the specific characteristics of the fault.

Added the reference Carlson *et al.*,1994.

**Referee's comment**. Finally, the writing is poor. Ideas from dynamical systems theory are mangled and confused in a way that I would need an hour to unpick. Same with the English: too many examples to deal with. Let me just mention the second sentence of

the abstract:

"The mathematical springblock model is generated by considering the Dieterich-Ruinas's friction law and the Stribeck's effect." would work much better if it were something like "The mathematical spring-block model includes Dieterich-Ruina's friction law and Stribeck's effect" or "The mathematical spring-block model includes the Dieterich-Ruina friction law and the Stribeck effect." As it stands, the sentence incorrectly conflates two nouns, uses an awkward construction, uses the wrong possessive and misspells a name (Ruina, not Ruinas).

**Author's replay**. We use the basic theory of dynamic systems, necessary to show the point that concerns us: to propose an upper limit (through the necessary condition for stability), which marks the difference in oscillatory behavior, when this condition is satisfied, and when it fails. This study is a first proposal of a limit for the area of slow earthquakes, obtained from an earthquakes model. From here, we can go deeper theoretically, and apply it in simulations with real data, as far as posible. We are aware that the analysis can go deeper into mathematical theory, but we are also aware that the journal is mainly directed to geophysician and seismologist, so we try to use a language more in line with other authors who have analyzed dynamical systems of the mechanism of earthquakes with mass-spring systems. We hope that with the observations you have indicated, this article had improved on the structure and clarification of what is being investigated.

**Author's changes in manuscript**:

Typos, grammar and spelling mistakes have been revised and corrected:

All "et al" corrected to "*et al*".

Page 1, line 2:"Ruinas's" corrected to "Ruina's". Page 1, line 2: the sentence "The mathematical springblock model is generated by considering the Dieterich-Ruinas's friction law and the Stribeck's effect" was changed and corrected to "The mathematical spring-block model includes Dieterich-Ruina's friction law and Stribeck's effect".

Page 4, line 2: "relatives" corrected to "relative", line 11: "velocity function" corrected to "velocity dependent", line 17: "increase" corrected to "increasing", "decrease" corrected to "decreasing"

Page 5, Line 18: "taking account" corrected to "taking into account", lines 19-20: "third order system differential equations" corrected to "first order differential equation system".

Page 6, line 11: "has the components" corrected to "is given by"; Page 6, line 23, and page 7, line 15, Page 9, Line 10, Page 11: Table 1, caption: "polynomial characteristic" corrected to "characteristic polynomial".

Page 6, line 22: element 12 of Jacobian matrix is "1" corrected to "0",

Page 7, line 15: "Descarte's" corrected to Descartes'. Page 13, line 8, Page 16, line 13: "longitude" corrected to "length". Page 7, line 24: "conjugate complex eigenvalues" corrected to "complex conjugate eigenvalue"; Page 7, line 22: "is given a sufficient condition to stability" corrected to "a sufficient condition for stability is given".

Page 10, line 5: "has two conjugate complex eigenvalues" corrected to "has two complex conjugate eigenvalues", Page 11, line 1: "asymptotical" corrected to "asymptotic"; page 11 line 5: capital letter "J" was changed in "jacobian".

Page 15, line 7: "necessary condition for stability. The" corrected to "necessary condition for stability, the" Page 17, line 26: "stablished" corrected to "established". Figure 1, caption: "doted" corrected to "dashed". Figure 2, caption: "de" corrected to "the".

Other corrections:

Page 4, line 3: The variable $x$ assigned for the displacement of the block was replaced by the variable $y$.

Page 5, lines 3-8: The paragraph was clarified, rephrased, and relocated after

Equation (4); added the term "$\beta_1$" to equation (2), it could be 0 if it is considered in Dieterich-Ruina's friction law.

Page 5, line 21: "$F_0(v)$" was removed; page 5, line 27: added and modified "$\alpha = \{\alpha_1, \alpha_2, \alpha_3\}$; $\alpha_{1,2} = \frac{L\beta_2}{v_0^2 M}$, $\alpha_3 = \frac{L\beta_3}{v_0 M}$. The external force is $\hat{\tau}(\hat{t}) = \hat{c}\sin(\hat{w}\hat{t})$, where $\hat{c} = \frac{L}{v_0^2}$ and $\hat{w} = \frac{Lw}{v_0}$". Page 5, line 28: added the term "$\alpha_1$" to equation (6).

Page 6, line 3: We added the text "The function $f(x)$ on the right-hand side of Eq. (5) defines a mapping $f : \mathbb{R}^3 \to \mathbb{R}^3$. This mapping defines a vector field on $\mathbb{R}^3$. Thus, the system given by Eq. (5) induces in phase space $\mathbb{R}^3$ the flow $(\varphi^t), t \in \mathbb{R}$ such that each forward trajectory of the initial point $x_0 = x(t = 0)$ is the set $\{x(t) = \varphi^t(x_0) : t \geq 0\}$".

Page 6, line 12: added the term $\alpha_1$ in $\eta$; page 6, line 8: added $\hat{\tau}(\hat{t}) := \tau(t)$, $\hat{\omega} := \omega$, $\hat{c} := c$

page 6, line 18: "mean that every solution of the system $(\theta, u, v)$" corrected to "*i. e.*, every solution of the system $\varphi^t(x_0) = (\theta(t), u(t), v(t))$",

Page 6, lines 20-21: "where $f(x)$ is the vectorial field or right side" corrected to "where $f(x) = (f_1, f_2, f_3)$ is the vector field given by right-hand side", line 21: "with $\tau(t) = 0$; and" corrected to "with $\tau(t) = 0$; $(x_1, x_2, x_3) = (\theta, u, v)$; and".

Page 7, line 2: Equation (11) was modified to $\frac{Mv_0}{L} + \frac{A}{v_0} + \beta_2\mu e^{-\mu v_0} < \beta_3$; Page 7, line 14: "$Re(\lambda_i) \geq 0$ for one or more eigenvalues of $D_f(x^\star)$" corrected to "at least one eigenvalues of $D_f(x^\star)$ is positive, *i.e.* $Re(\lambda_i) \geq 0$".

Page 17, line 20: (Ruina, 1983) was changed by (Daub and Carlson, 2008), last one was introduced in the references section.

Please also note the supplement to this comment:
http://www.nonlin-processes-geophys-discuss.net/npg-2016-60/npg-2016-60-AC3-supplement.pdf

**Supplement:**

[Figure]

(a) $\Pi_1=(0.25,0.8,1.8)$ $\omega=0$

(b) $\Pi_2=(0.25,0.8,.5)$ $\omega=0$

(c) $\Pi_3=(0.25,0.8,3)$ $\omega=.1$

(d) $\Pi_4=(0.25,0.8,.5)$ $\omega=.1$

(e) $\Pi_1$ $\omega=0$

(f) $\Pi_2$ $\omega=0$

(g) $\Pi_3$ $\omega=.1$

(h) $\Pi_4$ $\omega=.1$

---

## Author Comment (AC4) · 23 May 2017

Dear Referee Num. 2;
I made a mistake in my last reply. The new section included is "3.4 Type of Hopf bifurcation" instead of "3.3.3 Numerical features from the Hopf bifurcation analysis". The figure of supplement was not included in the manuscript, but it is available as supplementary information. I am apologized for the inconveniences caused. Thank you for your comments of the paper.

<hr>

---

## Author Response (AR1)

Dear Dr. Newman,

The manuscript "An upper limit for slow earthquakes zone: self-oscillatory behavior through the Hopf bifurcation mechanism from a model of spring-block under lubricated surfaces" by V. Castellanos-Rodríguez, E. Campos-Cantón, R. Barboza-Gudiño, and R. Femat to be considered for publication at Nonlinear Processes in Geophysics has been modified according to the comments and suggestions of the anonymous referees. Below You can find a summary of changes made to the old version of the manuscript. Thank you for give us the opportunity to publish our research results.

Sincerely,

The authors

**Summary of changes made to the manuscript NPGD-2016-60**.
**General**:

- We have made an effort to correct typos, grammar and spelling mistakes.

- Comments raised by the reviewers have been attended (see the reply to specific comments raised by reviewers).

- We clarified the ideas of the manuscript by rephrasing and relocating some paragraphs.

- Some equations were relocated and/or modified in a way that did not change the results.

- We added comments and introductions in some sections.

- A brief new subsection was added.

- Some comments were added in the discussion section and conclusion.

- The labels of the graphs were changed to the normal letters and font times new roman.

- Acknowledgements were modified.

- Some references were added, and mistakes corrected.

**Note**: All pages and lines refer to the old manuscript.

1. **General changes.**
   All "et al" corrected to "*et al*", and "i.e." corrected to "*i. e.*". Spaces were placed where required.

2. **Abstract**
   Page 1, line 2: the sentence "*The mathematical springblock model is generated by considering the Dieterich-Ruinas's friction law and the Stribeck's effect*" was changed and corrected to "*The mathematical spring-block model includes Dieterich-Ruina's friction law and Stribeck's effect*".

3. **Section I: Introduction.**
   Page 1, line 20: "of this paper" changed to "of the study presented in this paper". Page 2: Figure 1, caption: "doted" corrected to "dashed".

   Page 2, line 17: "stablish" corrected to "establish".

   Page 3, line 17: "Complex oscillatory behavior was observed in these cases, at surrounding of transition" corrected to "At surrounding of transition, complex oscillatory behavior was observed in these cases"

   Page 4, line 2: "relatives" corrected to "relative". Figure 2, caption: "de" corrected to "the".

   Page 4, lines 2, 3: The variable $x$ assigned for the displacement of the block was replaced by the variable $y$.

4. **Section 2.1: The model.**
   Page 4, line 11: "velocity function" corrected to "velocity dependent".

   Page 4, line 17: "increase" corrected to "increasing", "decrease" corrected to "decreasing".

   Page 5, lines 3-8: The paragraph was clarified, rephrased, and relocated after equation (4); for a more general form, we added the term "$\beta_1$" to equation (2). $beta_1$ could be 0 if the Coulomb friction is considered to be complete in Dieterich-Ruina's friction law, otherwise it takes other values.

Page 5, Line 18: "taking account" corrected to "taking into account".

Page 5, lines 19-20: "third order system differential equations" corrected to "first order differential equation system".

Page 5, line 21: "$F_0(v)$" was removed.

Page 5, line 27: added and modified "$\alpha = \{\alpha_1, \alpha_2, \alpha_3\}$; $\alpha_{1,2} = \frac{L\beta_2}{v_0^2 M}$, $\alpha_3 = \frac{L\beta_3}{v_0 M}$. The external force is $\hat{\tau}(\hat{t}) = \hat{c}\sin(\hat{w}\hat{t})$, where $\hat{c} = \frac{L}{v_0^2}$ and $\hat{w} = \frac{Lw}{v_0}$".

Page 5, line 28: added the term "$\alpha_1$" to equation (6).

Page 6, line 3: We added the text "The function $f(x)$ on the right-hand side of Eq. (5) defines a mapping $f : R^3 \to R^3$. This mapping defines a vector field on $R^3$. Thus, the system given by Eq. (5) induces in phase space $R^3$ the flow $(\varphi^t), t \in R$ such that each forward trajectory of the initial point $x_0 = x(t = 0)$ is the set $\{x(t) = \varphi^t(x_0) : t \geq 0\}$".

page 6, line 8: added $\hat{\tau}(\hat{t}) := \tau(t)$, $\hat{\omega} := \omega$, $\hat{c} := c$.

5. **Section 2.2: Stationary Solution at equilibrium point.**
Page 6, line 11: after "system (5)" add "with $\tau(t) = 0$", "has the components" corrected to "is given by".

Page 6, line 12: added the term $\alpha_1$ within the parenthesis of $\eta$.

Page 6, line 13: add "where" before "$\eta$".

Page 6, line 15: We added a clarifying text which explain why the linearized system is analyzed: In the spring-block model, the logarithmic term in the Dieterich-Ruina's friction law has introduced greater difficulty to solve the problem. Due to the nonlinear term, analytic integration has not been possible, and even numerical solutions present challenges because of the logarithmic term (Erickson *et al.*, 2008). The linearized system analysis is very useful to describe some features of the nonlinear system about steady state solution (Gu *et al.*, 1984; Shkoller and Minster, 1997; Erickson *et al.*, 2008).

page 6, line 18: "mean that every solution of the system $(\theta, u, v)$" corrected to "*i. e.*, every solution of the system $\varphi^t(x_0) = (\theta(t), u(t), v(t))$".

Page 6, lines 20-21: "where $f(x)$ is the vectorial field or right side" corrected to "where $f(x) = (f_1, f_2, f_3)$ is the vector field given by right-hand side", line 21: "with $\tau(t) = 0$; and" corrected to "with $\tau(t) = 0$; $(x_1, x_2, x_3) = (\theta, u, v)$; and".

Page 6, line 22: element (1,2) of Jacobian matrix is "1" corrected to "0" (This is typography mistake).

Page 6, line 23: "polynomial characteristic" corrected to "characteristic polynomial".

Page 7, line 2: Equation (11) was modified to $\frac{Mv_0}{L} + \frac{A}{v_0} + \beta_2\mu e^{-\mu v_0} < \beta_3$.

6. **Section 3: Oscillatory Behavior.** Page 7, line 11: added "We use the full nonlinear term in the numerical simulations in Section 3.2 and 3.3".

7. **Section 3.1: Analysis of Eigenvalues.**
Page 7, line 14: "$Re(\lambda_i) \geq 0$ for one or more eigenvalues of $D_f(x^\star)$" corrected to "at least one eigenvalues of $D_f(x^\star)$ is positive, *i.e.* $Re(\lambda_i) \geq 0$".

Page 7, line 15: "Descarte's" corrected to Descartes', "polynomial characteristic" corrected to "characteristic polynomial".

Page 7, line 22: "is given a sufficient condition to stability" corrected to "a sufficient condition for stability is given".

Page 7, line 24: "conjugate complex eigenvalues" corrected to "complex conjugate eigenvalue".

Other changes in Page 7: The Routh Hurwitz criteria have been relocated in this section. From page 8, lines 7 and 8 have been relocated in page 7, line 14; followed by lines 1-8 of page 11. Figure 5 was relocated in this section. Page 7, lines 20-22 are inserted and rephrased into one of the conditions for the Routh Hurwitz criteria.

8. **Section 3.2: A Hopf Bifurcation, Oscillatory Range (OR) and the Self-sustained Oscillations Region (SSO).**

Page 9, Line 10, and page 11, caption Table 1: "polynomial characteristic" corrected to "characteristic polynomial".

Page 10, line 5: "has two conjugate complex eigenvalues" corrected to "has two complex conjugate eigenvalues".

Page 11, line 1: "asymptotical" corrected to "asymptotic".

page 11 line 5: capital letter "J" was changed in "jacobian".

Page 11, lines 1-8 and Figure 5 have been relocated on page 7.

9. **Section 3.3: The System Under Forcing Conditions.**
Page 12, line 9: We added a brief introduction: This section aims to numerically describe the oscillatory behavior within and outside the range proposed for the SSO region (Eq. (21), page 10), under forcing and non-forced conditions. We want to prove numerically that the proposed upper limit determines the changes in oscillatory behavior, below and above this. For more theoretical background into the theory of periodically forced systems near a point of Hopf bifurcation, see Zhang and Golubitski (2011) and references therein.

Page 12, line 11: "$\alpha = \{0.2, 0.1\}$" corrected to "$\alpha = \{0.2, 0.2, 0.1\}$".

Page 12, Figure 6: The $x$ and $y$ labels have been changed to normal letters and font times new roman.

10. **Section 3.3.1: Bifurcation Diagram for Unperturbed System.**
Page 13, line 8: "longitude" corrected to "length".

Page 13, line 12: "are nearest" changed to "is nearest".

Page 13, Figure 7: The $y$ label "$u(t)$" changed to "$u_{lmax}(t)$", font times new roman.

11. **Section 3.3.2: Bifurcation Diagram for Perturbed System.**
Page 14, and page 15: Figures 9 and 10, caption: "$\alpha = \{0.2, 0.1\}$" corrected to "$\alpha = \{0.2, 0.2, 0.1\}$".

Page 14, Figures 8 and 9; page 15, Figure 10: The $y$ label "$u(t)$" changed to "$u_{lmax}(t)$". Normal letters, font times new roman.

Page 15, line 7: "necessary condition for stability. The" corrected to "necessary condition for stability, the".

Page 15, line 16: "it increases maintaining" corrected to "increases while maintaining".

12. **Section 3.4: Type of Hopf bifurcation.**
This is a brief new section about the type of Hopf bifurcation that could occur in the system in the parameter range that is handled in the article (see author's replay, page C5 and C6, interactive comment):

In terms of the flow in phase space, a supercritical Hopf bifurcation occurs when a stable spiral changes into an unstable spiral surrounded by nearly eliptical limit cycle (Strogatz, 1994). A subcritical Hopf bifurcation occurs when small perturbation can lead to either decaying oscillations due to a stable equilibrium or a jump to a large sustained oscillations in the system due to an unstable limit cycle. For the analysis of the bifurcation type, the main challenge is the numerical stiffness, due to the nonlinear logarithmic term.

The set of parameters $\Pi$ does no cross the Hopf bifurcation if $\gamma > \gamma_{HB}$. Small disturbances decay after ringing for a while and stable spiral is observed. The block and the driver plate are moving at constant rate $v = 1$, and the relative position is $\eta$. On the other hand for $\gamma < \gamma_{HB}$, the parameter values cross the Hopf bifurcation. The equilibrium state loses stability and unstable spiral is observed. This type of bifurcation is expected for smooth, non-catastrophic changes. The slow earthquakes are almost imperceptible because the displacement rate is very low compared to ordinary earthquakes and they are generated for parameter values around the critical value of nucleation. Hopf bifurcation is supercritical within the proposed limits for unforced system. To find chaotic behavior or strange attractors with the non-forced system it is necessary to vary epsilon very far (Erickson *et al.*, 2008) from the value of the Hopf bifurcation that we are analyzing.

However, Kostić *et al.* (2013) have found chaotic behavior for small values of $\Pi$ by introducing time delay in the friction term. They have found two types of Hopf bifurcation depending on the variation of the

time delay. Similarly, by introducing the external force $\tau(t)$ a subcritical Hopf Bifurcation could be given for some critical $\tau(t)$ and slight variation of the $\varepsilon$ and $\xi$ parameters. Disturbances do not allow the system to remain at an equilibrium point resulting in continuous oscillations or chaos. For the case when the set of parameters $\Pi$ crosses the Hopf bifurcation, continuous oscillations were found in both displacement and velocity only by varying the bifurcation parameter. Determining critical values of $\Pi$ and $\tau(t)$ requires more concrete study.

13. **Section 4: Discussion and Conclusions**

Page 16, line 13: "longitude" corrected to "length".

Page 16, line 23: "By other hand" changed to "On the other hand".

Page 17, line 20: "(Ruina, 1983)" changed to "(Daub and Carlson, 2008)"; "by other hand" corrected to "on the other hand".

Page 17, line 15: "He" changed to "Scholz (1998)"

Page 17, line 26: "stablished" corrected to "established".

Page 17, line 31: We added paragraphs: Although this investigation is more related to the proposal of a formal pattern in the study of SSEs, and with a first approximation of the upper limit of the transition zone, this is considered as a preliminary study in order to be applied to the real seismogenic regions. However, the parameters considered for slow earthquakes are still being studied through observations, experiments, and by means of simulations, but there is still not something precise.

The study of SSEs in Cascadia (Watkins *et al.*, 2015) indicates a possible link between the observational and experimental data with the parameters involved in the most of models of earthquake's physic coupled to the Dieterich-Ruina's friction law. The slip amount of SSEs is in cm order but the average slip amount of smaller events are unknown. The effective normal stress in the range of 3-9 MPa produce fault slip consistent with some observed SSEs, $B - A$ is in the range (0.0015 to 0.003) of the slow slip section. At the top of the slow slip section $B - A$ is 0.003 and 0.001 at the base, $A \approx 0.02$, $L$ is in the range 1-50 $\mu$m (real $L$ is unknown), the rate of convergence (10 a 50 mm/year) represents the range of convergence rates of subduction zones where SSEs

are observed with GPS. These parameters could vary depending on the region that SEEs occur. Further, the critical value $K_c = (A - B)\sigma/L$ depend on $L$; viscosity=0.1 (nondimensional) has been used in earthquake models (Carlson *et al.*, 1994), but the estimation of the real viscosity depends on the region.

The proposed upper limit for the SSEs zone includes the fluids and oscillation frequency (and consequently, $L$), through $\psi$. They might be introduced into the simulations and experiments in order to see which are the implications over the recurrence times, duration and velocity of SSEs in real seismogenic regions. A final step would be using scaling laws for SSEs to determine the real values of parameters included either experimental and/or simulation data, such as the stiffness $K_c$ and viscosity, take into account the specific characteristics of the fault.

14. **Acknowledges**

The acknowledges have been modified:

This study was supported by CONACyT (support 44731), the Departments of Applied Mathematics and Applied Geosciences of Instituto Potosino de Investigación Científica y Tecnológica (IPICYT), and the Instituto de Geología, Universidad Autónoma de San luis Potosí, México. E. Campos Cantón acknowledges the CONACYT financial support for sabbatical at Department of Mathematics, University of Houston. He would also like to thank the University of Houston for his sabbatical support and to Professor Matthew Nicol for allowing him to work together closely and his valuable discussions on dynamical systems. The authors also acknowledge technical support from Irwin A. Díaz-Díaz.

15. **References.**

Page 19: Avrutin *et al.* (2014) has been withdrawn.

The following references have been added:
Carlson *et al.* (2014).

Daub and Carlson (2008).

Kostić *et al.* (2013).

Shkoller and Minster (1997).

Zhang and Golubitsky (2011).

[revised manuscript text omitted]

where the Stribeck's effect from Eq. (1) is given now by

20  $$F_0(v) = \beta_1 + \beta_2 e^{-\mu(v)} + \beta_3 v. \tag{4}$$

Note that the slider block velocity is relative to velocity of driver plate. From the second equation of the system (3), we infer that the block will oscillate with respect to the position of the plate. This equation tells us whether the oscillator is to the right or to the left with respect to the driver plate. Although the direction of the displacement of the system is always forward; if $v - v_0 > 0$, *i.e.*, if $v > v_0$, eventually the oscillator will be more advanced than the plate. On the other hand, if $v - v_0 < 0$, *i.e.*,
25  $v < v_0$ eventually the oscillator will be to the left of the plate. One of the objectives when introducing the function $sign(v - v_0)$ in the Eq. (1) is indicate this effect; assigning value 1 for the first case, -1 for the second case, and zero otherwise. We eliminate the $sign$ function because this effect is already considered in the second equation of the system (3).

Defining new variables $\hat{\theta}$, $\hat{v}$, $\hat{u}$ and $\hat{t}$ (Erickson *et al.*, 2008): $\theta/A = \hat{\theta}$, $v/v_0 = \hat{v}$, $u/L = \hat{u}$, $t(v_0/L) = \hat{t}$, the dimensionless system is given by the following equations

$$
\begin{aligned}
\dot{\hat{\theta}} &= -\hat{v}[\hat{\theta} + (1+\varepsilon)\ln(\hat{v})], \\
\dot{\hat{u}} &= \hat{v} - 1, \\
\dot{\hat{v}} &= -\gamma^2[\hat{u} + (1/\xi)(\hat{\theta} + \ln(\hat{v}))] - \alpha F_0(\hat{v}) + \hat{\tau}(\hat{t}),
\end{aligned} \tag{5}
$$

where $\alpha = \{\alpha_1, \alpha_2, \alpha_3\}$; $\alpha_{1,2} = \frac{L\beta_2}{v_0^2 M}$, $\alpha_3 = \frac{L\beta_3}{v_0 M}$ . The external force is $\hat{\tau}(\hat{t}) = \hat{c}\sin(\hat{w}\hat{t})$, where $\hat{c} = \frac{L}{v_0^2}$ and $\hat{w} = \frac{Lw}{v_0}$. The
5  frictional parameters $\alpha$ are associated with frictional coefficients from the Stribeck's effect and

$$
\alpha F_0(\hat{v}) = \alpha_1 + \alpha_2 e^{-\hat{\mu}\hat{v}} + \alpha_3 \hat{v}, \qquad \hat{\mu} = v_0 \mu. \tag{6}
$$

$x = (\hat{\theta}, \hat{u}, \hat{v})$ is the dimensionless state variable, $\hat{\theta}$ stands for the measurement of contact with asperities from the Dieterich-Ruina friction law; $\hat{u}$ is the dimensionless relative displacement between the block and the upper plate and $\hat{v} > 0$ is the dimensionless velocity of the block. The function $f(x)$ on the right-hand side of Eq. (5) defines a mapping $f : \mathbb{R}^3 \to \mathbb{R}^3$. This
10  mapping defines a vector field on $\mathbb{R}^3$. Thus, the system given by Eq. (5) induces in phase space $\mathbb{R}^3$ the flow $(\varphi^t)$, $t \in \mathbb{R}$ such that each forward trajectory of the initial point $x_0 = x(t=0)$ is the set $\{x(t) = \varphi^t(x_0) : t \geq 0\}$.

Parameters $\Pi = (\varepsilon, \xi, \gamma)$ are given as follows: $\varepsilon = (B-A)/A$; $\xi = kL/A$ and $\gamma = \sqrt{(k/M)}(L/v_0)$, associated with stress drop during displacement, deformation and the oscillation frequency, respectively. The Equation (5) is referred to the system of Dieterich-Ruina-Stribeck (DR-S) where $\tau(\hat{t}) = [0, 0, \hat{\tau}(\hat{t})]^T$ is the periodical and deterministic external force $\hat{\tau}(t) = \hat{c}\sin(\hat{\omega}t)$,
15  here $\hat{\omega}$ is the angular frequency. We named unperturbed system when $\hat{\tau}(t) = 0$ and perturbed system otherwise. We will denote $(\hat{\theta}, \hat{u}, \hat{v}) := (\theta, u, v)$, $\hat{\tau}(\hat{t}) := \tau(t)$, $\hat{\omega} := \omega$, $\hat{c} := c$, and $\hat{\mu} := \mu$.

**2.2 Stationary solution at equilibrium point**

The stationary solution $x = x^\star$ of the system (5) with $\tau(t) = 0$ is given by
20  $$
x^\star = (\theta^\star, u^\star, v^\star) = (0, \eta, 1), \qquad u^\star = \eta = (\alpha_1 + \alpha_2 e^{-\mu} + \alpha_3)/\gamma^2, \qquad \gamma > 0, \tag{7}
$$

where $\eta$ corresponds to the relative position of the single slider block. At $x^\star$ the plate and the block have the same velocity and the measure of the asperities contact is zero. Note that $\theta^\star$ and $v^\star$ do not depend on $\Pi$ but $u^\star$ depends on frequency oscillation $\gamma$ (consequently on $kL$) and frictional constants of Stribeck's effect, both are associated with the energy dissipation.

In the spring-block model, the logarithmic term in the Dieterich-Ruina's friction law has introduced greater difficulty to
25  solve the problem. Due to the nonlinear term, analytic integration has not been possible, and even numerical solutions present challenges because of the logarithmic term (Erickson *et al.*, 2008). The linearized system analysis is very useful to describe some features of the nonlinear system about steady state solution (Gu *et al.*, 1984; Shkoller and Minster, 1997; Erickson *et al.*, 2008). The local and asymptotic stability of $x^\star$ is analyzed with the indirect method of Lyapunov that consists in the analysis of the eigenvalues of the Jacobian matrix from the linearized system of Eq. (5) around the stationary solution (Khalil, 1996). Let

$x^\star$ be locally asymptotically stable, *i. e.*, every solution of the system $\varphi^t(x_0) = (\theta(t), u(t), v(t))$, starting near of the stationary solution, it remains at the surrounding of $x^\star$ all the time, and eventually the solution converges to $x^\star$ (convergence to frictional stability). Let us denote the Jacobian matrix as $D_f(x^\star) = (\partial f_i(x)/\partial x_j)\,|_{x^\star}$, for $i, j = 1, 2, 3$, where $f(x) = (f_1, f_2, f_3)$ is the vector field given by the right-hand side of (5) with $\tau(t) = 0$; $(x_1, x_2, x_3) = (\theta, u, v)$; and let $\lambda_i$ be the eigenvalues of $D_f(x^\star)$:

$$\quad D_f(x^\star) = \begin{pmatrix} -1 & 0 & -(1+\varepsilon) \\ 0 & 0 & 1 \\ -\gamma^2/\xi & -\gamma^2 & -\gamma^2/\xi - \phi, \end{pmatrix} \tag{8}$$

where $\phi = \alpha_2 \mu e^{-\mu} - \alpha_3$. The characteristic polynomial of (8) is given by:

$$P(\lambda) = a_0 \lambda^3 + a_1 \lambda^2 + a_2 \lambda + a_3, \tag{9}$$

whose coefficients are in terms of seismic parameters $\Pi$, and frictional coefficients $\alpha$ and $\mu$.

$$a_0 = 1; \qquad a_1 = 1 + \gamma^2/\xi + \phi; \qquad a_2 = \gamma^2(1 - \varepsilon/\xi) + \phi; \qquad a_3 = \gamma^2. \tag{10}$$

10     The dynamical system of earthquakes is a naturally dissipative phenomena due to this feature the dissipativity condition of the stationary solution is required. Thus locally the system is dissipative at $x^\star$ if $a_1 = -\text{Trace} D_f(x^\star) = \sum_{i=1}^{3} \lambda_i < 0$ that is true under the condition:

$$\frac{Mv_0}{L} + \frac{A}{v_0} + \beta_2 \mu e^{-\mu v_0} < \beta_3; \tag{11}$$

Equation (11) comes directly from $a_1 < 0$ and the values of $\gamma$, $\varepsilon$, $\xi$, and $\phi$. The Equation (11) is the necessary condition for
15   the system to be sub-damped , and oscillations can be observed; moreover due to $a_3 = -\det D_f(x^\star) < 0$, $x^\star$ is hyperbolic. Through the analysis of the eigenvalues of $D_f(x^\star)$ we will explain what implies a hyperbolic equilibrium point related to oscillatory behavior.

**3   Oscillatory Behavior**

The earthquakes dynamics is a nonlinear oscillatory phenomenon (Gu *et al.*, 1984; De Sousa Vieira, 1995; Levin, 1996;
20   Chelidze *et al.*, 2005; Maloney and Robbins, 2007; Erickson *et al.*, 2008; Dragoni and Santini, 2010; Castellanos-Rodríguez and Femat, 2013; Amendola and Dragoni, 2013; Abe and Kato, 2014); where the nonlinear complex behavior is attributable to the friction forces. The analysis of the oscillatory behavior is explored in this section. We use the full nonlinear term in the numerical simulation in Section 3.2 and 3.3.

**3.1   Analysis of Eigenvalues**

25   The equilibrium point $x^\star$ is locally asymptotically stable if the real part of all the eigenvalues is negative, *i.e.* $Re(\lambda_i) < 0$, and it is unstable if at least one eigenvalue of $D_f(x^\star)$ is positive, *i.e.* $Re(\lambda_i) \geq 0$. We are interested in the type of hyperbolic

stationary solution $x^\star = (0, \eta, 1)$: it has a stable manifold; *i.e.*, $Re\{\lambda_i\} < 0$, $Im\{\lambda_i\} = 0$, and a unstable manifold that generates oscillations in a plane, *i.e.*, $Re\{\lambda_k\} > 0$, $Im\{\lambda_k\} \neq 0$ (Campos-Cantón *et al.*, 2010). A sufficient condition for local and asymptotic stability comes from the Routh-Hurwitz criterion, *i. e.*, the sufficient conditions to ensure that Jacobian matrix (8) has three eigenvalues with negative real part is that the coefficients of the characteristic polynomial holds the inequalities:

$$a_0, a_1, a_2, a_3 > 0 \qquad a_1 a_2 - a_0 a_3 > 0. \tag{12}$$

Note that the first inequality holds: $a_0, a_3 > 0$, and $a_1 > 0$ because $\phi < 1$ and $\Pi > 0$; if $sign(a_2) > 0$ we deduce

$$\varepsilon < \xi\psi; \qquad \psi = 1 + \phi/\gamma^2; \tag{13}$$

as a necessary condition for stability but it is not sufficient. The sufficient condition for stability comes from the second inequality of (12) and $a_2 > 0$ as follow

[revised manuscript text omitted]

**3.4 Type of Hopf bifurcation**

In terms of the flow in phase space, a supercritical Hopf bifurcation occurs when a stable spiral changes into an unstable spiral surrounded by nearly eliptical limit cycle (Strogatz, 1994). A subcritical Hopf bifurcation occurs when small perturbation can lead to either decaying oscillations due to a stable equilibrium or a jump to a large sustained oscillations in the system due to an unstable limit cycle. For the analysis of the bifurcation type, the main challenge is the numerical stiffness, due to the nonlinear logarithmic term. The set of parameters $\Pi$ does no cross the Hopf bifurcation if $\gamma > \gamma_{HB}$. Small disturbances decay after ringing for a while and stable spiral is observed. The block and the driver plate are moving at constant rate $v = 1$, and the relative position is $\eta$. On the other hand for $\gamma < \gamma_{HB}$, the parameter values cross the Hopf bifurcation. The equilibrium state loses stability and unstable spiral is observed. This type of bifurcation is expected for smooth, non-catastrophic changes. The slow earthquakes are almost imperceptible because the displacement rate is very low compared to ordinary earthquakes and they are generated for parameter values around the critical value of nucleation. Hopf bifurcation is supercritical within the proposed limits for unforced system. To find chaotic behavior or strange attractors with the non-forced system it is necessary to vary epsilon very far (Erickson *et al.*, 2008) from the value of the Hopf bifurcation that we are analyzing. However, Kostić *et al.* (2013) have found chaotic behavior for small values of $\Pi$ by introducing time delay in the friction term. They have found two types of Hopf bifurcation depending on the variation of the time delay. Similarly, by introducing the external force $\tau(t)$ a subcritical Hopf Bifurcation could be given for some critical $\tau(t)$ and slight variation of the $\varepsilon$ and $\xi$ parameters. Disturbances do not allow the system to remain at an equilibrium point resulting in continuous oscillations or chaos. For the case when the set of parameters $\Pi$ crosses the Hopf bifurcation, continuous oscillations were found in both displacement and velocity only by varying the bifurcation parameter. Determining critical values of $\Pi$ and $\tau(t)$ requires more concrete study.

[revised manuscript text omitted]

Although this investigation is more related to the proposal of a formal pattern in the study of SSEs, and with a first approximation of the upper limit of the transition zone, this is considered as a preliminary study in order to be applied to the real seismogenic regions. However, the parameters considered for slow earthquakes are still being studied through observations, experiments, and by means of simulations, but there is still not something precise.

25  The study of SSEs in Cascadia (Watkins *et al.*, 2015) indicates a possible link between the observational and experimental data with the parameters involved in the most of models of earthquake's physic coupled to the Dieterich-Ruina's friction law. The slip amount of SSEs is in cm order but the average slip amount of smaller events are unknown. The effective normal stress in the range of 3-9 MPa produce fault slip consistent with some observed SSEs, $B-A$ is in the range (0.0015 to 0.003) of the slow slip section. At the top of the slow slip section $B-A$ is 0.003 and 0.001 at the base, $A \approx 0.02$, $L$ is in the range 1-50

30  $\mu$m (real $L$ is unknown), the rate of convergence (10 a 50 mm/year) represents the range of convergence rates of subduction zones where SSEs are observed with GPS. These parameters could vary depending on the region that SEEs occur. Further, the

critical value $K_c = (A - B)\sigma/L$ depend on $L$; viscosity=0.1 (nondimensional) has been used in earthquake models (Carlson *et al.*, 1994), but the estimation of the real viscosity depends on the region.

The proposed upper limit for the SSEs zone includes the fluids and oscillation frequency (and consequently $L$), through $\psi$. They might be introduced into the simulations and experiments in order to see which are the implications over the recurrence times, duration and velocity of SSEs in real seismogenic regions. A final step would be using scaling laws for SSEs to determine the real values of parameters included either experimental and/or simulation data, such as the stiffness $K_c$ and viscosity, take into account the specific characteristics of the fault.

*Competing interests.* The authors declare that they have no conflict of interests.

*Acknowledgements.* This study was supported by CONACyT (support 44731), the Departments of Applied Mathematics and Applied Geosciences of Instituto Potosino de Investigación Científica y Tecnológica (IPICYT), and the Instituto de Geología, Universidad Autónoma de San luis Potosí, México. E. Campos Cantón acknowledges the CONACYT financial support for sabbatical at Department of Mathematics, University of Houston. He would also like to thank the University of Houston for his sabbatical support and to Professor Matthew Nicol for allowing him to work together closely and his valuable discussions on dynamical systems. The authors also acknowledge technical support from Irwin A. Díaz-Díaz.